# Concept-Guided Tokenization: Closing the Gap Between Reconstruction and Generation

Yunqiao Yang[1,2]   Haokun Lin[2,3]   Guanzhong Wu[4]   Ying Wei[4]

## Abstract

Recent advances in image generation have been largely driven by image tokenization, which compresses raw pixels into compact latent representations. While existing tokenizers excel at preserving low-level visual details through reconstruction-based training, they often lack explicit semantic guidance, which limits their ability to capture semantically structured representations and thus hinders their performance on downstream tasks like image generation. To overcome this limitation, we propose a novel tokenization framework that incorporates high-level semantics through two key innovations: (1) a text-integrated encoder that jointly processes images and textual descriptions to produce semantically enriched latent representations, and (2) a concept-guided training objective that leverages sparse autoencoders to decompose pre-trained vision-language model features to a semantic concept space, employing sparse and disentangled concept indices for guidance. Our approach achieves strong alignment with semantic concepts, maintaining high reconstruction fidelity with an rFID of 1.39 on ImageNet, while achieving a gFID of 2.65 on the class-conditional image generation task and 10.73 on the text-to-image generation task. By infusing high-level semantic structures into low-level visual fidelity, our method bridges the reconstruction-generation divide and drives generative modeling as a powerful foundation. The code is available at https://github.com/hustyyq/ConceptTok.

[1]College of Computing and Data Science, Nanyang Technological University [2]Department of Computer Science, City University of Hong Kong [3]Institute of Automation, Chinese Academy of Sciences [4]College of Computer Science and Technology, Zhejiang University. Correspondence to: Ying Wei <ying.wei@zju.edu.cn>.

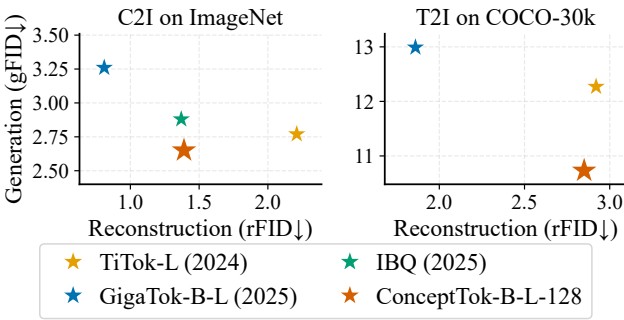

*Figure 1.* Reconstruction-generation trade-off of different tokenizers on class-to-image (C2I) and text-to-image (T2I) generation.

## 1. Introduction

In recent years, image generation has achieved remarkable progress, with diffusion (Rombach et al., 2022; Peebles & Xie, 2023; Hatamizadeh et al., 2024; Shin et al., 2025) and autoregressive (Sun et al., 2024; Li et al., 2024; Tian et al., 2024; Li et al., 2025a) models achieving high-quality synthesis. A key enabler of this progress is image tokenization, which compresses raw pixels into a compact latent space (Yu et al., 2024a; Xiong et al., 2025a). These latent representations, whether continuous (Kingma & Welling, 2014; Li et al., 2024) or discrete (Van Den Oord et al., 2017; Yu et al., 2022), provide an expressive yet computationally efficient alternative to the high-dimensional image space. Tokenization enables generative models to operate directly in the latent domain, simultaneously improving efficiency and synthesis fidelity (Zha et al., 2025; Kim et al., 2025) and thus establishing it as an important component of image generation systems.

Reconstruction-based training (Yu et al., 2024a; Zha et al., 2025; Kim et al., 2025) serves as a primary objective for learning visual tokenizers, as it effectively preserves low-level image details. However, while achieving strong reconstruction performance, this approach often lacks high-level semantic guidance (Qu et al., 2025; Wu et al., 2025b; Zhao et al., 2025) so as to poorly generalize to downstream generation tasks with limited quality of generated images (Xiong et al., 2025b; Hansen-Estruch et al., 2025), well known as *reconstruction-generation trade-off*. To overcome this, ex-

isting methods (Qu et al., 2025; Chen et al., 2025b; Xiong et al., 2025b) typically align tokenizer features with high-level representations from pre-trained vision models such as CLIP (Radford et al., 2021) or DINOv2 (Oquab et al., 2024), or encourage text-image alignment using paired captions (Ge et al., 2024; Liang et al., 2024; Wu et al., 2025b). Yet, direct alignment with pre-trained feature representations introduces both optimization and generalization challenges. First, the high dimensionality of these features makes alignment a difficult regression task, where the curse of dimensionality flattens distance metrics and weakens gradients. Second, the semantic entanglement nature of pre-trained representations makes them a less qualified and potentially biased source for alignment. Given that individual dimensions in pre-trained representations conflate multiple concepts (*e.g.*, "bird beak" and "leaves") (Gandelsman et al., 2025; Lim et al., 2025), alignment is more likely to emphasize dominant features (*e.g.*, "leaves"). As a result, fine-grained semantics crucial for bird generation (*e.g.*, "bird beak") may be overlooked.

To overcome this limitation, we propose a *Concept*-guided *Tok*enizer (ConceptTok), introducing two key innovations. First, instead of aligning with the dense pre-trained features, we disentangle them into a semantic concept space via sparse autoencoders (SAEs) (Gao et al., 2025a; Lim et al., 2025) and take the top-$K$ activated concept indices as alignment signals (Tack et al., 2026). This conceptual alignment source enjoys three key merits: (1) *Sparsity*. We introduce the TopK SAE (Gao et al., 2025a) to transform the feature representations of a pre-trained vision-language model (SigLIP (Zhai et al., 2023)) into a small set of $K$ activated concept indices, focusing the alignment loss on this sparse set and yielding sharper, more consistent gradients. (2) *Low dimensionality*. Alignment operates in a $K$-dimensional concept space rather than the original dense feature space, simplifying optimization and mitigating noise from irrelevant dimensions. (3) *Disentanglement*. The tokenizer learns to predict fine-grained semantic concepts (*e.g.*, "bird beak") rather than imitating entangled features, reducing latent space complexity and promoting compositional generalization (*e.g.*, "bird beak" transferring from jays to crows), which improves downstream generation.

Second, we further endow the latent space with semantic structure by integrating textual information as an additional input modality. Recognizing that textual descriptions naturally capture higher-level abstractions (Zha et al., 2025; Kim et al., 2025), our tokenizer encoder processes both images and their corresponding text captions to produce a semantically compact latent representation. Unlike prior works that also condition on text at the decoder (de-tokenization) stage (Zha et al., 2025; Kim et al., 2025), which likely biases text-to-image generation without improving the latent representations themselves, our method integrates textual

information only into the encoder. The introduction of textual modality further warrants downstream generation tasks, through structurally coherent and semantically rich latent space representations.

Our contributions are summarized as follows:

- By aligning ConceptTok representations with sparse concept indices extracted via an SAE and integrating textual conditioning exclusively at the encoder rather than the decoder, ConceptTok learns a semantically structured latent space that better balances reconstruction fidelity and downstream generation performance.

- Our approach achieves strong performance on the C2I (ImageNet (Russakovsky et al., 2015)) and T2I (COCO-30k (Lin et al., 2014)) benchmarks, obtaining 1.39 rFID / 2.65 gFID on ImageNet and 2.85 rFID / 10.73 gFID on COCO-30k, thereby demonstrating a strong reconstruction-generation trade-off (Fig. 1).

## 2. Related Work

**Image Tokenizers** compress high-resolution images into compact tokens within a latent space, which can be either discrete (Van Den Oord et al., 2017; Razavi et al., 2019; Esser et al., 2021; Yu et al., 2022) or continuous (Kingma & Welling, 2014; Li et al., 2024). This transformation enables downstream tasks to operate directly in the compressed latent space, substantially improving efficiency for image generation (Lee et al., 2022; Chang et al., 2022; Rombach et al., 2022) and understanding (Ning et al., 2023). While reconstruction-based training effectively preserves low-level image details (Yu et al., 2024b), it overlooks semantic structure, limiting generalization to downstream image generation (Yu et al., 2025; Xiong et al., 2025b; Kim et al., 2025).

To address this limitation, recent methods instill explicit semantic guidance into the tokenization process to improve the generalization and effectiveness of the learned latent representations: (1) some approaches align latent features with vision representations extracted from pre-trained models (Qu et al., 2025; Yao et al., 2025; Chen et al., 2025b; Xiong et al., 2025b; Zhang et al., 2025b), such as CLIP (Radford et al., 2021) or DINOv2 (Oquab et al., 2024); (2) others enforce text-image alignment, encouraging latent features to capture semantics consistent with the images' textual descriptions (Ge et al., 2024; Liang et al., 2024; Wu et al., 2025b); (3) yet others directly map images into the token space of a frozen large language model, treating text tokens as the codebook for image representation (Yu et al., 2023; Zhu et al., 2024). However, these strategies align the entire representations, facing the dual challenges of high-dimensional optimization and semantic entanglement. In contrast, our work introduces concept guidance that first projects pre-trained representations into a concept space,

providing sparse, low-dimensional, and disentangled alignment signals.

Closely related work also conditions on the text at the de-tokenization stage or at both stages to guide image reconstruction (Zha et al., 2025; Kim et al., 2025). Yet, their performance improvements largely stem from text conditioning applied at the de-tokenization stage. This design externalizes semantic information to the text, leaving the latent space less self-contained as a standalone representation for downstream image generation tasks.

**Concept-Oriented Tokenizers** seek to discover structural or semantic patterns directly from images. Some approaches partition an image into an adaptive number of regions through segmentation and encode each region into a token (Wang et al., 2024; Wu et al., 2025a; Chen et al., 2025a; Yin et al., 2025). However, the discovered "concepts" typically correspond to concrete pixel regions rather than higher-level semantics, such as style (Karras et al., 2019) or lighting (Bhattad et al., 2024), and thus struggle to capture semantic abstractions crucial for image generation. Other methods represent an image as a set of disentangled visual concept tokens, with each token responding to a distinct visual concept learned solely through reconstruction (Locatello et al., 2020; Yang et al., 2022). These methods generally assume that images consist of a small number of independent and disentangled concepts and are mainly applied to synthetic datasets, limiting their generalization to natural images with complex semantics (Wang et al., 2024).

**Sparse Autoencoders** (SAEs) enforce sparsity in the latent space of an autoencoder by restricting the number of active latent dimensions (Lee et al., 2006). This constraint promotes the learning of disentangled and compact representations that often correspond to coherent semantic concepts (Huben et al., 2024). This capability has led to the broad adoption of SAEs across both natural language processing (Gao et al., 2025a; Karvonen et al., 2025) and computer vision (Lim et al., 2025; Zaigrajew et al., 2025), enabling applications of model interpretability (Huben et al., 2024), model output steering (Lieberum et al., 2024), and LLM pre-training (Tack et al., 2026).

## 3. Preliminaries

**1D Tokenizer** We build our tokenizer upon TiTok (Yu et al., 2024b), a vision Transformer (ViT) (Dosovitskiy et al., 2021) based one-dimensional vector-quantized (VQ) (Esser et al., 2021) model. Its Transformer architecture enables it to process arbitrary token sequences, making it suited for jointly modeling image and text inputs. Given an RGB image $I \in \mathbb{R}^{H \times W \times 3}$, where $H$ and $W$ denote the image height and width, respectively, TiTok partitions the image into non-overlapping patches and linearly projects them into

patch tokens $P \in \mathbb{R}^{(\frac{H}{f} \times \frac{W}{f}) \times D}$. Here, $f$ is the patch size, $(\frac{H}{f} \times \frac{W}{f})$ is the number of patches, and $D$ is the patch embedding dimension. The patch tokens are concatenated with learnable latent tokens $L \in \mathbb{R}^{N \times D}$, where $N$ is the number of learnable latent tokens. The ViT encoder $\mathrm{Enc}$ processes this sequence to produce the latent representations:

$$[\_; Z_{1\mathrm{D}}] = \mathrm{Enc}([P; L]), \tag{1}$$

where $[\cdot; \cdot]$ denotes concatenation along the token sequence dimension, the output $\_$ corresponding to the patch tokens $P$ is discarded, and $Z_{1\mathrm{D}} \in \mathbb{R}^{N \times D}$ corresponding to the learnable tokens $L$ serves as the compressed representations used in subsequent steps. We then apply vector quantization (Esser et al., 2021) to $Z_{1\mathrm{D}}$ to obtain discrete latent codes. The ViT decoder $\mathrm{Dec}$ takes the quantized tokens $\mathrm{Quant}(Z_{1\mathrm{D}})$ and a new set of learnable patch tokens $P' \in \mathbb{R}^{(\frac{H}{f} \times \frac{W}{f}) \times D}$ to reconstruct the image:

$$[\_; \widehat{I}] = \mathrm{Dec}([\mathrm{Quant}(Z_{1\mathrm{D}}); P']), \tag{2}$$

where $\widehat{I}$ denotes the reconstructed image, while the first $N$ outputs are discarded.

**Sparse Autoencoder** An SAE automatically maps the latent feature representations of a pre-trained model into a semantic concept space; given an image, it extracts a set of concept indices within this space. Given the hidden state $h \in \mathbb{R}^{d_h}$ extracted from an image by a pre-trained model, the SAE maps $h$ into high-dimensional activations $c \in \mathbb{R}^{d_c}$ through a linear encoder, and reconstructs the original input via a linear decoder (Lee et al., 2006). Sparsity constraints are imposed on $c$ to produce compact and interpretable representations, where each active index is encouraged to align with a semantically meaningful concept (Huben et al., 2024; Lim et al., 2025). In this work, we adopt a TopK SAE (Makhzani & Frey, 2014; Gao et al., 2025a) that restricts each input to a small, fixed number of active concept indices, which serve as stable and reliable alignment signals.

Formally, the SAE consists of a linear encoder $\mathbf{W}_1 \in \mathbb{R}^{d_h \times d_c}$ and a linear decoder $\mathbf{W}_2 \in \mathbb{R}^{d_c \times d_h}$, with bias terms $b_1 \in \mathbb{R}^{d_c}$ and $b_2 \in \mathbb{R}^{d_h}$. Given a hidden state $h$, the encoding, sparsification, and reconstruction steps follow:

$$c' = \mathbf{W}_1^\top (h - b_2) + b_1, \tag{3}$$

$$c = \mathrm{ReLU}\left(\mathrm{TopK}(c')\right), \tag{4}$$

$$\hat{h} = \mathbf{W}_2^\top c + b_2. \tag{5}$$

The training objective with respect to $\mathbf{W}_1, \mathbf{W}_2, b_1, b_2$ is to minimize the reconstruction error, *i.e.*,

$$\ell_{\mathrm{SAE}} = \|h - \hat{h}\|_2^2. \tag{6}$$

Here, $c'$ denotes the pre-activation concept vector, $\mathrm{TopK}(\cdot)$ retains only the $K$ largest activations while setting the rest

to zero, followed by $\mathrm{ReLU}$ activation, and $c$ is the resulting sparse concept sets. The reconstruction $\hat{h}$ is obtained by decoding $c$. By enforcing Top-$K$ sparsity, the SAE isolates the most salient dimensions in $c'$, each indexing a semantic concept within the image.

## 4. Concept-Guided Tokenization

In this section, we present the framework of our ConceptTok, which incorporates text conditioning and concept guidance. The overview of ConceptTok is illustrated in Fig. 2.

### 4.1. Text-Integrated Tokenizer

Most existing methods rely exclusively on image inputs, overlooking accompanying text descriptions that could serve as complementary semantic information (Zha et al., 2025). Although some prior approaches incorporate text conditioning (Zha et al., 2025; Kim et al., 2025), they primarily focus on text-guided image reconstruction at the de-tokenization stage. As a result, such designs tend to bias the model toward text-to-image generation rather than optimizing the compressed latent representations. Consequently, the latent representations remain weakly grounded in semantics and are less effective as self-contained representations for image generation models. In contrast, our method integrates texts only into the encoder, explicitly encouraging more semantically meaningful and structurally coherent latent representations that support downstream generation.

Given an image and its corresponding text caption (the construction of which is described in Sec. 5.1), our tokenizer accepts both modalities as input. For the text, we first extract semantic embeddings using a pre-trained CLIP text encoder, followed by a linear projection to align the feature dimension with the patch tokens $P$. This produces text tokens $T \in \mathbb{R}^{T \times D}$, where $T$ denotes the text sequence length. The tokenizer encoder then concatenates the text tokens, patch tokens, and learnable latent tokens $L$, and compresses them into the latent representation:

$$[\_; \_; Z_{1\mathrm{D}}] = \mathrm{Enc}([T; P; L]),  \qquad (7)$$

where $Z_{1\mathrm{D}}$ associated with $L$ is preserved for subsequent processing, while the outputs corresponding to $T$ and $P$ are discarded. The reconstruction stage follows the same formulation as in Eq. (2).

### 4.2. Concept-Guided Training

Prior tokenizer training (Yu et al., 2024a; Zha et al., 2025; Kim et al., 2025) typically employs reconstruction-based losses, collectively denoted as $\ell_{\mathrm{VQGAN}}$, which commonly include $\ell_2$ reconstruction loss, perceptual loss (Johnson et al., 2016), adversarial loss with a PatchGAN discriminator (Isola et al., 2017), and LeCam regularization (Tseng et al., 2021) for stabilizing adversarial training. In addition, a VQ codebook loss (Esser et al., 2021) optimizes the discrete latent codebook. While these objectives facilitate high-fidelity image reconstruction, they predominantly emphasize low-level visual details and often fail to encourage semantically meaningful latent representations (Xiong et al., 2025b; Lin et al., 2025). Recent approaches (Qu et al., 2025; Xiong et al., 2025b) attempt to address this by aligning latent features with those from pre-trained models for visual understanding (e.g., SigLIP), thereby leveraging the richer semantic structure encoded in these models.

Unfortunately, directly aligning with entire pre-trained feature representations still suffers from both optimization and generalization challenges. First, features from pre-trained models are high-dimensional (e.g., 768 for CLIP/B or DINOv2/B), making such alignment a high-dimensional regression problem where the curse of dimensionality flattens distance metrics and weakens optimization gradients. Second, holistic feature alignment operates on entangled embeddings, where each dimension encodes mixed semantic concepts (e.g., "bird beak" and "bird feet") (Lim et al., 2025). Consequently, gradients are spread thinly across many entangled dimensions, likely biasing the model towards dominant but less informative features (e.g., "leaves") while diluting the fine-grained signals critical for generation.

To address these limitations, we introduce a concept-guided training objective that encourages the latent $Z_{1\mathrm{D}}$ to capture fine-grained concept semantics. Rather than aligning with dense pre-trained features, we project pre-trained features into a semantic concept space via a TopK SAE (Gao et al., 2025a) and treat the top-$K$ (e.g., $K = 128$) activated concept indices as alignment signals. Such sparse, low-dimensional, and disentangled supervision yields clearer alignment signals and eases optimization. More importantly, the tokenizer learns to predict disentangled semantic concepts rather than imitate entangled embeddings, thereby reducing latent space complexity, promoting compositional transfer, and facilitating generalization in downstream generation.

Specifically, we utilize a pre-trained vision-language model (e.g., SigLIP (Zhai et al., 2023)) that captures and aligns rich visual-textual semantics to discover semantic concepts. We train a TopK SAE on features of its vision encoder using the LLaVA-NeXT dataset (Liu et al., 2024) that provides high-quality aligned image-text pairs with rich semantic correspondence. The SAE produces a concept space from which we extract the top-$K$ activated concept indices. Let $\mathcal{I} = \{i_1, i_2, \ldots, i_K\}$ denote the set of indices corresponding to the top-$K$ entries in the SAE's activations $c$. To inject this semantic information into our tokenizer, we propose a concept loss to explicitly encourage the latent space accurately predictive of the activated indices in the concept

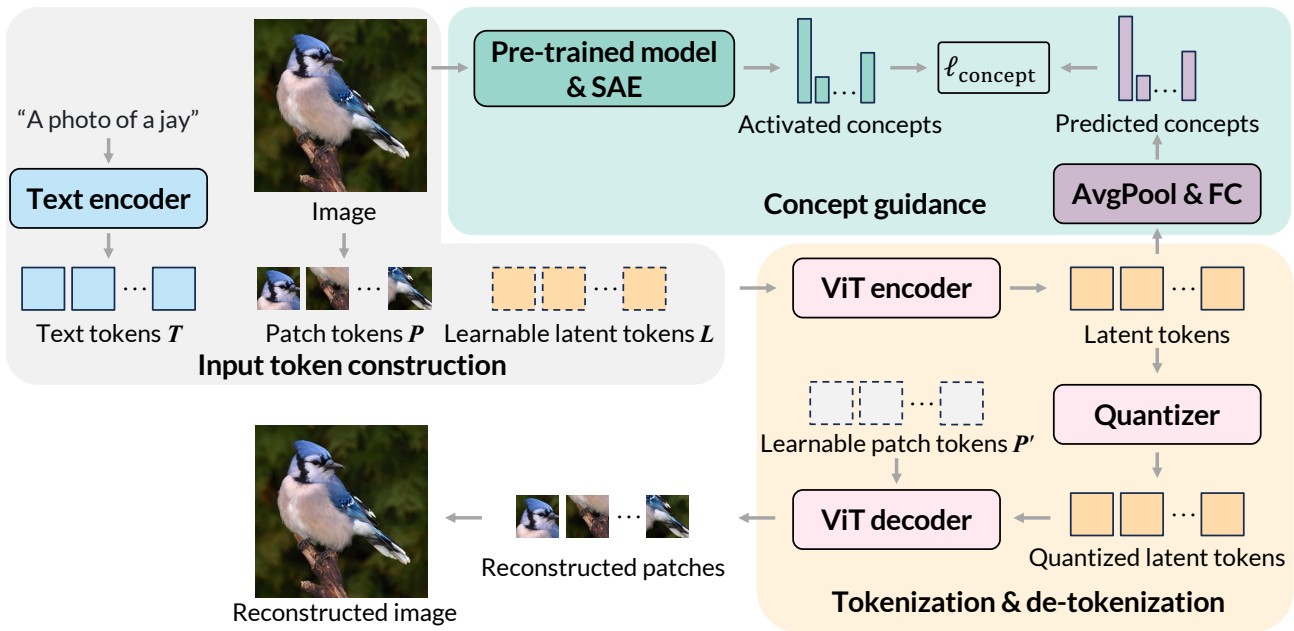

*Figure 2.* Overview of ConceptTok. The tokenizer encoder (ViT) jointly encodes text tokens, patch tokens, and learnable latent tokens to produce latent representations. The model vector-quantizes the latent tokens and then feeds them, together with learnable patch tokens, into the ViT decoder to reconstruct the image. During training, an SAE extracts sparse activated concept indices from pre-trained image features, and the tokenizer is guided to predict these activated concepts.

space. We obtain the predicted concept vector from the latent features by first average pooling the latent sequence $Z_{1D}$ and then applying a fully-connected (FC) layer $\phi$, *i.e.*,

$$z = \phi\left(\mathrm{AvgPool}(Z_{1D})\right). \tag{8}$$

We compute the concept loss as,

$$\ell_{\mathrm{concept}} = \frac{1}{K} \sum_{i \in \mathcal{I}} \mathrm{CE}(z, i), \tag{9}$$

where CE denotes the cross-entropy loss. A smaller CE loss contributes to a more semantically structured and conceptually aligned latent space. The complete training objective combines the concept loss with conventional reconstruction-based losses:

$$\ell_{\mathrm{total}} = \ell_{\mathrm{VQGAN}} + \lambda \cdot \ell_{\mathrm{concept}}, \tag{10}$$

where $\lambda$ is a weighting hyperparameter. This overall objective ensures that the tokenizer learns to strike a balance between input image reconstruction and high-level semantic concept learning, which contributes to more effective latent representations for downstream vision tasks.

## 5. Experiments

In this section, we first describe the experimental setups and then compare ConceptTok with existing tokenizers on ImageNet and COCO-30k benchmarks. Next, we analyze

whether the learned latent space aligns with the SAE-derived concept space and compare concept alignment with dense feature alignment. Finally, we study the contribution of each ConceptTok component.

### 5.1. Experimental Setups

**Tokenization Models** We implement our tokenizers using ViT-based architectures, specifically ViT/S, ViT/B, and ViT/L, which serve as encoder and decoder components. For example, a tokenizer designated as "B-L" uses a ViT/B encoder and a ViT/L decoder. Following Kim et al. (2025), we use input images with a resolution of $256 \times 256$, which we partition into $16 \times 16$ patches, yielding 256 patch tokens. To support different compression ratios, we vary the number of latent tokens as $N \in \{32, 64, 128\}$. For the vector-quantized model, we use a codebook with size 8,192 and dimensionality 64, consistent with Kim et al. (2025).

**Image Generative Models** Following Xiong et al. (2025b), we evaluate the applicability of our tokenizers to class-conditional image generation (C2I) using two variants of LlamaGen (Sun et al., 2024): LlamaGen-B and LlamaGen-XL. Class conditioning is implemented via learnable embeddings (Esser et al., 2021), which serve as prefixed class tokens indicating the specific ImageNet (Russakovsky et al., 2015) class. Starting from the class token, the generative model autoregressively predicts a sequence of latent tokens whose length matches the number of latent tokens of

*Table 1.* Main results on ImageNet $256 \times 256$. Reconstruction performance is evaluated on ImageNet validation set, while generation performance follows the evaluation protocols in ADM (Dhariwal & Nichol, 2021). "Type" specifies the generative model family, where "Diff.", "AR" and "Mask." correspond to diffusion, autoregressive, and masked Transformer models, respectively. [†]: training with an entropy loss (Yu et al., 2024a) and online clustered codebook (Zheng & Vedaldi, 2023). [‡]: training with additional data beyond ImageNet. [⋆]: C2I generation without classifier-free guidance.

| Tokenizer | Param. | #Tokens | rFID↓ | rIS↑ | Generator | Param. | Type | gFID↓ | gIS↑ |
|---|---|---|---|---|---|---|---|---|---|
| **Continuous tokens** | | | | | | | | | |
| VAE (Rombach et al., 2022) | 55M | 4096 | 0.27 | – | LDM-4 (Rombach et al., 2022) | 400M | Diff. | 3.60 | – |
| SD-VAE (Ma et al., 2024) | 84M | 1024 | 0.62 | – | SiT-XL/2 (Ma et al., 2024) | 675M | Diff. | 2.06 | – |
| VA-VAE (Yao et al., 2025) | 70M | 256 | 0.28 | 205.6 | LightningDiT (Yao et al., 2025) | 675M | Diff. | 1.35 | 295.3 |
| VAE (Li et al., 2024) | 66M | 256 | 0.53 | – | MAR-H (Li et al., 2024) | 943M | AR+Diff. | 1.55 | 303.7 |
| **Discrete tokens** | | | | | | | | | |
| B-AE-d32 (Wang et al., 2023) | 66M | 256 | 1.69 | – | BiGR-XXL-d32 (Hao et al., 2025) | 1.5B | AR+Diff | 2.36 | 277.2 |
| VQGAN (Chang et al., 2022) | 66M | 256 | 2.28 | – | MaskGIT (Chang et al., 2022) | 227M | Mask. | 6.18⋆ | – |
| TiTok-B (Yu et al., 2024b) | 202M | 64 | 1.70 | – | MaskGIT-ViT (Chang et al., 2022) | 177M | Mask. | 2.48 | 214.7 |
| TiTok-L (Yu et al., 2024b) | 641M | 32 | 2.21 | – | MaskGIT-ViT (Chang et al., 2022) | 177M | Mask. | 2.77 | 199.8 |
| VAR-Tok (Tian et al., 2024) | 109M | 680 | 1.00[‡] | – | VAR-d24 (Tian et al., 2024) | 1.0B | VAR | 2.09 | 312.9 |
| VAR-Tok (Tian et al., 2024) | 109M | 680 | 1.00[‡] | – | VAR-d30 (Tian et al., 2024) | 2.0B | VAR | 1.92 | 323.1 |
| ImageFolder (Li et al., 2025a) | 176M | 286 | 0.80[‡] | – | ImageFolder-VAR (Li et al., 2025a) | 362M | VAR | 2.60 | 295.0 |
| ViT-VQGAN (Yu et al., 2022) | 64M | 1024 | 1.28 | 192.3 | VIM-Large (Yu et al., 2022) | 1.7B | AR | 4.17⋆ | 175.1⋆ |
| Open-MAGVIT2 (Luo et al., 2024) | 133M | 256 | 1.17 | – | Open-MAGVIT2-B (Luo et al., 2024) | 343M | AR | 3.08 | 258.3 |
| Open-MAGVIT2 (Luo et al., 2024) | 133M | 256 | 1.17 | – | Open-MAGVIT2-XL (Luo et al., 2024) | 1.5B | AR | 2.33 | 271.8 |
| IBQ (Shi et al., 2025) | 128M | 256 | 1.37 | – | IBQ-B (Shi et al., 2025) | 342M | AR | 2.88 | 254.7 |
| IBQ (Shi et al., 2025) | 128M | 256 | 1.37 | – | IBQ-XXL (Shi et al., 2025) | 2.1B | AR | 2.05 | 286.7 |
| LlamaGenTok (Sun et al., 2024) | 72M | 256 | 2.19 | – | LlamaGen-B (Sun et al., 2024) | 111M | AR | 5.46 | 193.6 |
| LlamaGenTok (Sun et al., 2024) | 72M | 256 | 2.19 | – | LlamaGen-XL (Sun et al., 2024) | 775M | AR | 3.39 | 227.1 |
| GigaTok-B-L (Xiong et al., 2025b) | 622M | 256 | 0.81 | – | LlamaGen-B (Sun et al., 2024) | 111M | AR | 3.26 | 221.0 |
| GigaTok-XL-XXL (Xiong et al., 2025b) | 2.9B | 256 | 0.79 | – | LlamaGen-B (Sun et al., 2024) | 111M | AR | 3.15 | 224.3 |
| ConceptTok-B-L | 533M | 128 | 2.38 | 285.8 | LlamaGen-B (Sun et al., 2024) | 111M | AR | 2.97 | 232.4 |
| ConceptTok-B-L[†] | 533M | 128 | 1.39 | 258.3 | LlamaGen-XL (Sun et al., 2024) | 775M | AR | 2.65 | 237.9 |
| ConceptTok-B-L | 533M | 128 | 2.38 | 285.8 | LlamaGen-XL (Sun et al., 2024) | 775M | AR | 2.37⋆ | 248.7⋆ |

the tokenizer. These predicted tokens are then decoded by the pre-trained tokenizer decoder to produce the final image.

We additionally evaluate text-to-image (T2I) generation using MaskGen-L (Kim et al., 2025). MaskGen generates images by iteratively predicting masked latent tokens conditioned on text inputs, providing a complementary evaluation setting to autoregressive generation.

**Training Setups** We train a TopK SAE to map image representations from a pre-trained SigLIP-B/16 (Zhai et al., 2023) into a semantic concept space. The SAE operates on the final-layer features of the SigLIP vision encoder, which are aligned with text embeddings and therefore intrinsically semantic. We train the SAE on the LLaVA-NeXT dataset (Liu et al., 2024), using a concept space dimensionality of $d_c = 24,576$ and the sparsity parameter of $K = 128$.

We train the tokenizers on the ImageNet training set (Russakovsky et al., 2015) at a resolution of $256 \times 256$. Since ImageNet does not provide captions, we construct class-descriptive text prompts using the template "*A photo of a {class name}*" (Kim et al., 2025). Training proceeds for 200 epochs with a learning rate of $10^{-4}$ and a cosine decay schedule (Loshchilov & Hutter, 2017). The trade-off parameter $\lambda$ is selected via grid search over $\{0.01, 0.1\}$ to balance reconstruction and concept-guidance objectives.

For downstream evaluation, we train the C2I LlamaGen frameworks on ImageNet for 300 epochs using the WSD scheduler (Hägele et al., 2024), with a base learning rate of $10^{-4}$ and a decay ratio of 0.2, following Xiong et al. (2025b). For T2I evaluation, we train MaskGen-L (Kim et al., 2025) on CC3M (Sharma et al., 2018) for 125 epochs with a learning rate of $4 \times 10^{-4}$ and a cosine decay schedule.

**Evaluation Metrics** We evaluate reconstruction quality using reconstruction Fréchet Inception Distance (rFID) (Heusel et al., 2017) and reconstruction Inception Score (rIS) (Salimans et al., 2016) on the ImageNet validation set, and rFID on COCO-30k (Lin et al., 2014; Li et al., 2025b), all at a resolution of $256 \times 256$. To assess downstream image generation performance, we train C2I generative models (*i.e.*, LlamaGen) on ImageNet using the learned tokenizers and report generation Fréchet Inception Distance (gFID) and generation Inception Score (gIS), following the evaluation protocol of ADM (Dhariwal & Nichol, 2021). We further train T2I masked generative models (*i.e.*, MaskGen-L) on CC3M and evaluate gFID on COCO-30k.

### 5.2. Main Results

We evaluate ConceptTok against state-of-the-art tokenizers

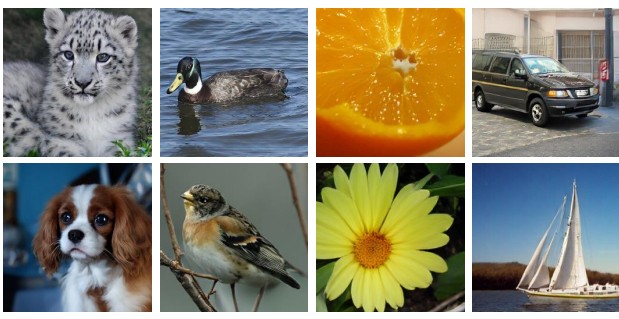

*Figure 3.* Class-conditional generation results from ConceptTok-B-L-128 using the LlamaGen-XL framework, producing high-fidelity images with fine-grained concept details.

*Table 2.* Comparison of reconstruction and T2I (MaskGen-L) generation performance of different tokenizers on COCO-30k.

| Tokenizer | rFID↓ | gFID↓ |
|---|---|---|
| TiTok-B-L-128 (Kim et al., 2025) | 2.92 | 12.27 |
| GigaTok-B-L-256 (Xiong et al., 2025b) | 1.86 | 12.99 |
| ConceptTok-B-L-128[†] | 2.85 | 10.73 |

*"Three stuffed bears hugging and sitting on a blue pillow."*

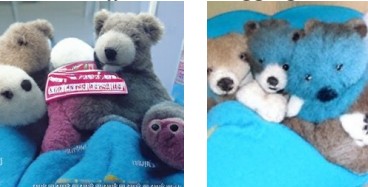 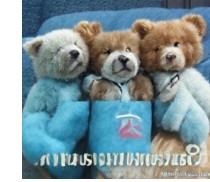

*"Two little birds sitting on top of a wedding cake."*

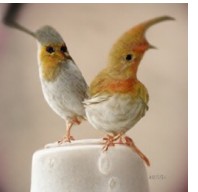 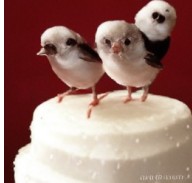 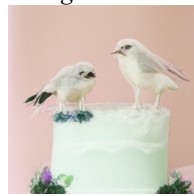

| TiTok | GigaTok | ConceptTok |
|---|---|---|

*Figure 4.* Qualitative comparisons of images generated from identical text prompts using different tokenizers. ConceptTok produces images with higher fidelity and better semantic alignment.

on ImageNet for reconstruction and C2I generation, and on COCO-30k for T2I generation. Among the compared methods, GigaTok (Xiong et al., 2025b) incorporates semantic guidance by aligning tokenizer features with those from a pre-trained model via cosine similarity maximization.

**Reconstruction**  ConceptTok achieves competitive reconstruction fidelity, as shown in Tab. 1. Specifically, ConceptTok-B-L-128[†] attains an rFID of 1.39 on ImageNet.

**C2I Generation**  For downstream C2I generation, the semantically structured latent space induced by ConceptTok enables strong generalization and consistently competitive performance. When paired with the LlamaGen-XL generator, ConceptTok-B-L-128 achieves a gFID of 2.37, which is comparable to significantly larger autoregressive generative models like Open-MAGVIT2-XL (2.33 gFID) and IBQ-XXL (2.05 gFID). Importantly, ConceptTok operates on sequences of only 128 latent tokens, which is half the length of the 256 tokens used by Open-MAGVIT2 and IBQ, and thus provides a superior trade-off between generation quality and inference efficiency. Under the same LlamaGen-B generator, ConceptTok-B-L-128 (533M, 2.97 gFID) not only outperforms the GigaTok-B-L (622M, 3.26 gFID) but also surpasses the much larger GigaTok-XL-XXL (2.9B, 3.15 gFID). These results demonstrate that concept-guided tokenizers produce more semantically structured latent representations than holistic feature alignment, leading to superior generalization in downstream image generation tasks.

As a qualitative demonstration, Fig. 3 presents C2I examples from our ConceptTok-B-L-128 paired with the LlamaGen-XL generator. The generated images exhibit semantically rich structures and fine visual details, corroborating the high quantitative performance of our method. We provide more C2I examples in Fig. 8 in the Appendix.

**T2I Generation**  As shown in Tab. 2, ConceptTok achieves substantially stronger T2I generation performance while maintaining reconstruction quality comparable to existing

tokenizers. This indicates that ConceptTok attains a more favorable reconstruction-generation trade-off, attributable to the semantically structured latent space, which generalizes well to complex textual prompts and enables robust transfer to text-conditioned image generation.

As a qualitative demonstration, Fig. 4 presents visual comparisons across different tokenizers under identical text prompts and MaskGen-L. ConceptTok generates images with higher fidelity and clearer alignment with the texts (*e.g.*, three bears or two birds). These results highlight ConceptTok's ability to preserve both fine-grained visual details and high-level semantics in T2I generation. We provide more T2I examples in Fig. 9 in the Appendix.

### 5.3. Concept Alignment Analysis

**Quantitative Concept Alignment**  We adopt the F1 score to quantitatively evaluate semantic alignment between our tokenizer and the SAE-derived concept indices. This metric measures the overlap between the top-$K$ concept indices identified by the pre-trained SigLIP SAE and those predicted by our tokenizer. We report results on three datasets: the ImageNet validation set, COCO-30k, and a T2I set generated by ConceptTok-B-L-128[†]. As shown in Tab. 3, all ConceptTok variants exhibit strong alignment on the ImageNet validation set, and maintain substantial alignment

*Table 3.* Concept alignment scores (F1 score) on ImageNet validation, COCO-30k, and the T2I sets. Results demonstrate strong alignment on the ImageNet validation set, effective generalization to COCO-30k, and preserved semantics in generated images.

| Tokenizer | ImageNet | COCO | T2I set |
| --- | --- | --- | --- |
| ConceptTok-B-B-64 | 0.601 | 0.508 | 0.445 |
| ConceptTok-B-L-64 | 0.618 | 0.542 | 0.465 |
| ConceptTok-B-L-128 | 0.623 | 0.542 | 0.458 |

*Table 4.* Ablation study of ConceptTok under the B-B-64 configuration. Both text conditioning (TC) and concept guidance (CG) improve performance, confirming each component's contribution.

| Component | rFID↓ | rIS↑ | gFID↓ | gIS↑ |
| --- | --- | --- | --- | --- |
| Baseline | 7.95 | 96.5 | 7.31 | 170.1 |
| + CG | 4.66 | 129.4 | 6.51 | 171.5 |
| + TC | 4.66 | 284.8 | 5.31 | 250.3 |
| + TC + CG | 3.84 | 276.0 | 4.13 | 245.8 |
| + TC + Cosine alignment | 4.30 | 263.9 | 4.84 | 212.6 |

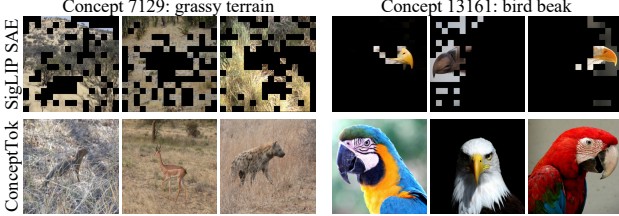

Concept 7129: grassy terrain          Concept 13161: bird beak

*Figure 5.* Concept alignment visualization. The top row highlights patches that activate a given SigLIP SAE-derived concept index. The bottom row presents images for which ConceptTok produces the highest responses for the corresponding concept index. The retrieved images consistently exhibit the corresponding semantics across diverse contexts, indicating strong concept alignment.

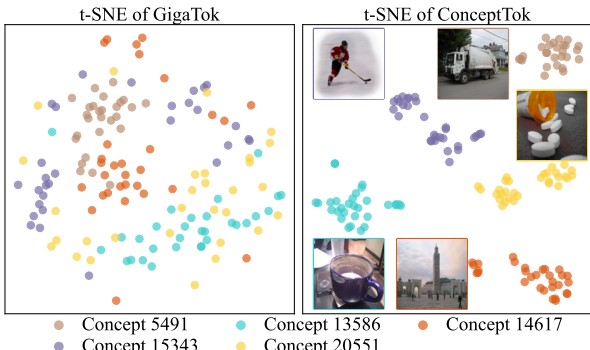

t-SNE of GigaTok          t-SNE of ConceptTok

- Concept 5491    - Concept 13586    - Concept 14617
- Concept 15343    - Concept 20551

*Figure 6.* Comparison of latent space structure via t-SNE. GigaTok embeddings are semantically entangled, while ConceptTok embeddings form discernible semantic clusters, demonstrating more concept-structured representations.

when generalized to COCO-30k. Even on the T2I-generated set, the tokenizers achieve an F1 score of approximately $0.45$, indicating that the generated images retain meaningful semantic concepts captured by the SAE.

**Qualitative Concept Alignment** We provide qualitative visualizations on ImageNet validation set to illustrate concept alignment. For each concept index identified by the SigLIP SAE, we retrieve images with the highest activations, and highlight the specific patches whose SigLIP representations activate the concept index while masking other regions (Lim et al., 2025). As shown in Fig. 5, the top row presents patch-level visualizations for SAE concept indices (*e.g.*, "grassy terrain" and "bird beak"), where the unmasked regions correspond to patches that activate each concept. The bottom row displays images for which our tokenizer's latent representations $Z_{1D}$ produce the highest scores for these same concept indices. Notably, the retrieved images consistently contain the corresponding semantics despite contextual variations, such as antelopes and leopards in grassy terrain. This consistency across contexts demonstrates that our tokenizer achieves fine-grained concept alignment with the pre-trained model. Additional examples are in Fig. 10 in the Appendix.

**Latent Space Analysis** We further analyze the structure of the learned latent feature space using t-SNE (Maaten & Hinton, 2008), as shown in Fig. 6, with more examples for SAE-derived concept indices provided in Fig. 11 in the Appendix. The results show that ConceptTok produces latent

representations that form distinct clusters corresponding to semantic concepts, indicating successful alignment of the representation space around a semantic concept space. This structured latent space reduces complexity and facilitates more effective downstream generative model training.

**Comparison with Holistic Feature Alignment** Following REPA (Yu et al., 2025), we measure the feature alignment between the tokenizer and the pre-trained model using CKNNA (Huh et al., 2024) on ImageNet. ConceptTok-B-B-64 achieves a CKNNA score of $0.48$, whereas replacing the concept guidance loss $\ell_{concept}$ with a holistic cosine similarity alignment (Xiong et al., 2025b) reduces it to $0.39$, indicating weaker representation alignment. This suggests that projecting features into a sparse, disentangled SAE-derived concept space is more effective than direct dense-feature cosine alignment. Consistently, Tab. 4 shows that substituting $\ell_{concept}$ with holistic cosine alignment also degrades reconstruction and generation performance, demonstrating that concept guidance better structures the latent space and improves generalization in image generation.

**Extension to Generator Feature Alignment** To further validate the benefit of concept alignment, we apply it to diffusion image generator feature alignment on the class-conditional ImageNet 256×256 benchmark. REPA (Yu et al., 2025) improves diffusion Transformer training by

*Table 5.* Diffusion generator feature alignment on C2I ImageNet $256 \times 256$. All generators are trained for 400K iterations under the same setup, changing only the auxiliary alignment objective.

| Model | Alignment | Trade-off | gFID↓ |
|---|---|---|---|
| SiT-L/2 (Ma et al., 2024) | None | 0.0 | 18.8 |
| SiT-L/2-Atten (Ma et al., 2024) | Attention matrix | 1.0 | 15.6 |
| SiT-L/2-REPA (Yu et al., 2025) | Dense feature | 0.5 | 9.7 |
| SiT-L/2-CG | Concept space | 0.1 | 9.2 |
| SiT-L/2-CG | Concept space | 0.2 | 8.7 |

adding an auxiliary feature alignment loss, which aligns intermediate denoising features with image representations extracted by DINOv2. Following REPA (Yu et al., 2025), we train SiT-L/2 (Ma et al., 2024) for 400K iterations and keep the pretrained model and training setup unchanged, modifying only the auxiliary alignment objective. We additionally compare with attention-matrix alignment (Gao et al., 2025b), which aligns the attention maps of the diffusion generator and the pretrained model. As shown in Tab. 5, attention alignment improves over vanilla SiT, reducing gFID from 18.8 to 15.6. Dense feature alignment, *i.e.*, REPA, further improves gFID to 9.7, while our SAE-derived concept guidance (CG) achieves the best result of 8.7. These suggest that concept-space alignment provides more effective semantic guidance than both attention-matrix alignment and holistic dense feature alignment, and that its benefit extends beyond tokenizer training to diffusion generator alignment.

## 5.4. Ablation and Further Analysis

**Component Ablation** We conduct an ablation study to assess the contribution of each component in ConceptTok on ImageNet. Quantitative results in Tab. 4 demonstrate a clear performance trend. Compared with the baseline, incorporating either concept guidance or text conditioning leads to substantial improvements. Combining both components further reduces the rFID to 3.84 and the gFID to 4.13. This progressive improvement confirms that both components are crucial for learning semantically structured representations that generalize effectively to image generation tasks.

**Comparison with TA-TiTok** We further compare ConceptTok with TA-TiTok-B-L-128 (Kim et al., 2025), a closely related text-aware tokenizer that incorporates textual information at the de-tokenization stage, as shown in Tab. 6. When class-informative captions are replaced by the generic prompt "A photo", TA-TiTok's ImageNet rFID increases from 1.53 to 2.64, while ConceptTok changes from 1.39 to 1.95, indicating that de-tokenizer stage conditioning is more sensitive to informative captions. Although TA-TiTok obtains better COCO reconstruction rFID, likely benefiting from its recaptioned DataComp (Gadre et al., 2023) training, ConceptTok achieves better COCO T2I gFID under the same 50-epoch MaskGen setting. These results support our

*Table 6.* Controlled comparison with TA-TiTok under the B-L-128 setting. For T2I generation, both methods use 50-epoch MaskGen training on CC3M. "class" denotes "*A photo of a {class name}*", and "generic" denotes "*A photo*".

| Tokenizer | ImageNet rFID↓ | | COCO-30k | |
|---|---|---|---|---|
| | class | generic | rFID↓ | T2I gFID↓ |
| TA-TiTok | 1.53 | 2.64 | 2.43 | 16.54 |
| ConceptTok[†] | 1.39 | 1.95 | 2.85 | 12.64 |

*Table 7.* Image-conditioned generation via inpainting on COCO. We use a center mask covering $40\%$ of the image width and height.

| Tokenizer | LPIPS↓ | SSIM↑ |
|---|---|---|
| TA-TiTok-B-L-128 | 0.264 | 0.829 |
| ConceptTok-B-L-128[†] | 0.210 | 0.847 |

design choice of encoder-stage text conditioning, which produces more self-contained and semantically enriched latent tokens for downstream generation.

**Image Inpainting** We further evaluate ConceptTok on image-conditioned generation via inpainting. Specifically, we train an inpainting ViT-B on CC3M to predict the tokenizer tokens of the original image from those of a masked image, and evaluate on COCO using a fixed center mask covering $40\%$ of the image width and height. We report the structural similarity index (SSIM) (Wang et al., 2004) and the learned perceptual image patch similarity (LPIPS) (Zhang et al., 2018) between the inpainted images and the tokenizer reconstructions of the original images. As shown in Tab. 7, ConceptTok outperforms TA-TiTok in both metrics, demonstrating that its latent tokens are more self-contained and effective for image-conditioned generation.

**Additional Ablations** We provide additional ablation studies on the number of latent tokens, tokenizer model scale, the trade-off parameter $\lambda$, and the choice of pretrained models in Appendix B.1.

## 6. Conclusion

We present ConceptTok, a novel tokenization framework that integrates text conditioning and concept guidance to improve the semantic structure of the latent space. The resulting representations demonstrate strong concept alignment, effectively closing the reconstruction-generation trade-off. We discuss limitations and future directions in Appendix D.

## Acknowledgements

This work was supported in part by grants from the National Natural Science Foundation of China (No. 62441236).

## Impact Statement

This work utilizes publicly available resources, including ImageNet, LLaVA-NeXT, the pre-trained SigLIP model, and the pre-trained CLIP model, all of which are widely adopted within the research community. We acknowledge that these resources may contain inherent knowledge that may be reflected in our tokenizer's outputs. We also recognize the risks associated with the misuse of image reconstruction and generation technologies, particularly in sensitive contexts. Consequently, we emphasize that deploying such technologies requires careful ethical consideration.

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

# A. More Implementation Details

## A.1. SAE Training Details

We use SigLIP-B/16 (Zhai et al., 2023) with an input resolution of $256 \times 256$, taking the final layer of the vision encoder, which has a feature dimension of 768. The SAE is trained with a learning rate of $4 \times 10^{-4}$ and a constant warm-up schedule with 500 warm-up steps (Lim et al., 2025). We train the SAE for 13,640 iterations with a batch size of 192 on the LLaVA-NeXT dataset (Liu et al., 2024), using ghost gradients for optimization (Lim et al., 2025).

The SAE is trained once as an auxiliary module and then frozen during tokenizer training. Its concept space can therefore be reused across different tokenizer variants and ablations. In our implementation, SAE training requires only 1.35 hours on 8 A100 GPUs. Thus, the additional SAE stage introduces a small one-time cost compared with the main tokenizer training.

## A.2. Tokenizer Training Details

We provide detailed training hyperparameters for our tokenizers in Tab. 8.

*Table 8.* Training hyperparameters.

| Hyperparameter | Value |
|---|---|
| $\ell_2$ loss weight | 1.0 |
| Quantizer loss weight | 1.0 |
| Concept loss weight | 0.01 / 0.1 |
| Adversarial loss weight | 0.1 |
| Discriminator starting epoch | 80 |
| Perceptual loss weight | 1.1 |
| Perceptual loss models | LPIPS VGG (Zhang et al., 2018) ConvNeXt Small (Liu et al., 2022) |
| LeCam weight | 0.001 |
| Entropy loss weight | 0 / 0.01 |
| Learning rate | $10^{-4}$ |
| Optimizer | AdamW (Loshchilov & Hutter, 2019) $(\beta_1 = 0.9, \beta_2 = 0.999)$ |
| Learning rate schedule | Cosine learning decay (Loshchilov & Hutter, 2017) |
| Weight decay | $10^{-4}$ |
| Training epochs | 200 |
| Batch size | 256 / 512 |

## A.3. Image Generation Details

**Classifier-free Guidance** The gFID of generative models can be significantly influenced by classifier-free guidance (CFG) (Ho & Salimans, 2022; Sun et al., 2024). To be consistent with previous work (Xiong et al., 2025b), we perform a grid search for the optimal CFG scale within the range 1.0 to 3.0 with step size 0.25. Specifically, we follow the approach in (Xiong et al., 2025b) where models generate the first $18\%$ of tokens without guidance (*i.e.*, CFG scale = 1.0) to encourage image diversity, after which CFG is applied to the remaining tokens to enhance visual quality.

# B. More Experimental Results

## B.1. Ablation Studies

In the ablation studies, all reconstruction results are evaluated on the ImageNet validation set, while all generation results are obtained on the C2I generation task on ImageNet using LlamaGen-B as the autoregressive image generator.

**Number of Latent Tokens** We analyze the impact of the number of latent tokens on tokenization performance. As shown in Tab. 9, increasing from 32 to 128 tokens reduces rFID from 7.52 to 2.38 and gFID from 6.19 to 2.97, indicating that a

*Table 9.* Ablation study on the number of latent tokens $N$.

| Tokenizer | rFID↓ | rIS↑ | gFID↓ | gIS↑ |
|---|---|---|---|---|
| B-L-32 | 7.52 | 332.1 | 6.19 | 297.7 |
| B-L-64 | 4.52 | 325.8 | 3.28 | 254.2 |
| B-L-128 | 2.38 | 285.8 | 2.97 | 232.4 |

*Table 10.* Ablation study on tokenizer model scale.

| Tokenizer | Param. | rFID↓ | rIS↑ | gFID↓ | gIS↑ |
|---|---|---|---|---|---|
| S-S-64 | 189M | 6.68 | 192.0 | 7.13 | 241.2 |
| B-B-64 | 316M | 3.84 | 276.0 | 4.13 | 245.8 |
| B-L-64 | 533M | 4.52 | 325.8 | 3.28 | 254.2 |

larger number of latent tokens enables the tokenizer to capture richer information.

**Tokenizer Model Scale**   We study the effect of tokenizer model scale by comparing different architectures with $64$ latent tokens, as shown in Tab. 10. Increasing model capacity consistently improves downstream generation performance, highlighting the importance of tokenizer scale.

**Trade-off Parameter** $\lambda$   We analyze the impact of the trade-off parameter $\lambda$ using the ConceptTok-B-B-64 configuration, as summarized in Tab. 11. Setting $\lambda = 0$ (*i.e.*, without concept guidance) yields inferior reconstruction and generation performance. Moderate values of $\lambda$ significantly improve both rFID and gFID, with $\lambda = 0.1$ achieving the best reconstruction and generation trade-off, highlighting the importance of concept guidance. In contrast, overly large values of $\lambda$ (*e.g.*, $\lambda = 0.5$) degrade performance, suggesting that excessive emphasis on concept guidance can hinder effective visual reconstruction.

*Table 11.* Ablation study on $\lambda$ using ConceptTok-B-B-64.

| $\lambda$ | rFID↓ | rIS↑ | gFID↓ | gIS↑ |
|---|---|---|---|---|
| 0.0 | 4.66 | 284.8 | 5.31 | 250.3 |
| 0.01 | 3.87 | 265.4 | 4.77 | 240.5 |
| 0.1 | 3.84 | 276.0 | 4.13 | 245.8 |
| 0.5 | 3.99 | 258.6 | 4.84 | 213.7 |

**Choice of Pre-trained Models**   We further analyze the impact of different pre-trained models using the ConceptTok-S-S-64 configuration. The results in Tab. 12 indicate that using CLIP/B and SigLIP/B achieve comparable performance across reconstruction and generation metrics, relatively insensitive to the specific choice of pre-trained model.

**Performance Trends**   Based on the preceding ablation studies on ConceptTok, we further summarize the relationship between reconstruction and generation performance in Fig. 7. Each point corresponds to a different ConceptTok variant. We observe a strong positive correlation between rFID and gFID, indicating that improved reconstruction fidelity consistently benefits downstream image generation, mitigating the reconstruction-generation trade-off (Xiong et al., 2025b). Moreover, generation performance improves steadily with increasing tokenizer model size, demonstrating favorable scalability.

### B.2. Effect of Richer Captions

Since ImageNet does not provide natural language captions, our main experiments use the class-template prompt "*A photo of a {class name}*". To examine whether richer textual descriptions further benefit ConceptTok, we additionally train ConceptTok-B-L-128[†] on recaptioned ImageNet [1], while keeping the tokenizer training recipe unchanged. As shown in Tab. 13, descriptive captions improve both reconstruction and downstream generation, reducing ImageNet rFID from 1.39 to

---

[1] https://huggingface.co/datasets/visual-layer/imagenet-1k-vl-enriched

*Table 12.* Ablation study on pre-trained models using ConceptTok-S-S-64.

| Pre-trained models | rFID↓ | rIS↑ | gFID↓ | gIS↑ |
|---|---|---|---|---|
| SigLIP/B | 6.68 | 192.0 | 7.13 | 241.2 |
| CLIP/B | 6.45 | 203.8 | 6.80 | 227.4 |

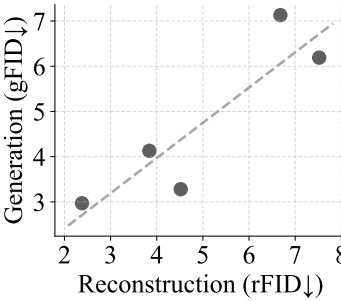 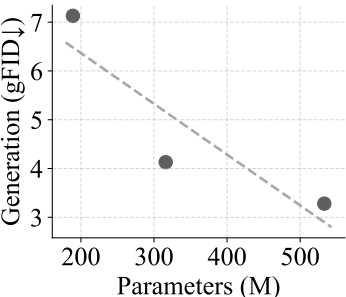

*Figure 7.* In ConceptTok, better reconstruction and larger tokenizers correlate with improved class-conditional image generation, alleviating the generation-reconstruction dilemma.

1.28, COCO rFID from 2.85 to 2.49, and COCO T2I gFID from 10.73 to 10.16. These results suggest that ConceptTok can benefit from more informative captions.

### B.3. More Visualizations

**C2I Visualizations**   We provide additional C2I examples to illustrate ConceptTok's effectiveness. Fig. 8 shows images generated by ConceptTok-B-L-128 using the LlamaGen-XL generator, across a diverse range of object categories.

**T2I Visualizations**   To complement the examples presented in the main paper, Fig. 9 presents additional T2I examples for four prompts. The comparison across different tokenizers, all using the MaskGen-L generator, highlights ConceptTok's superior ability to generate images with fine-grained visual details and accurate semantic alignment.

**Concept Alignment Visualization**   We present additional qualitative visualizations of concept alignment on the ImageNet validation set in Fig. 10. Following the same procedure as in the main paper, we visualize patch-level activations for concept indices identified by the SigLIP SAE and retrieve images with high latent scores produced by ConceptTok.

**SAE-derived Concept Index Visualization**   We present additional examples of SAE-derived concept indices in Fig. 6. To interpret the learned representations, we visualize concepts derived from the SAE by identifying images that yield the highest activation for each concept index, as shown in Fig. 11. Following Zhang et al. (2025a), we use a multimodal large language model (*e.g.*, Qwen2.5-VL (Bai et al., 2025)) to generate descriptive words for these concepts based on their corresponding image sets. Finally, two authors verify the appropriateness of the proposed concept words to ensure semantic consistency.

## C. Discussion with RAEs

Recent representation autoencoder (RAE) (Zheng et al., 2026) and RAEv2 (Singh et al., 2026) provide strong alternatives for building semantically meaningful latent spaces for image generation. RAEs typically reuse fixed, pre-trained representation encoders to define continuous semantic latents, while ConceptTok keeps a trainable tokenizer and introduces concept-level feature alignment. Our intuition also aligns with PS-VAE (Zhang et al., 2025b), which suggests that both semantic alignment and reconstruction objectives are important for making representation encoders suitable for generation. From this perspective, ConceptTok provides a structured concept-level alignment objective that can be combined with reconstruction-based tokenizer training. Moreover, concept-level alignment is not tied solely to tokenization; it can also serve as a general representation-alignment signal for diffusion generators, as shown in our generator-alignment experiments.

*Table 13.* Effect of descriptive captions using ConceptTok-B-L-128[†]. Richer captions improve both reconstruction and downstream generation.

| Training captions | ImageNet rFID↓ | COCO rFID↓ | COCO T2I gFID↓ |
|---|---|---|---|
| Class-template captions | 1.39 | 2.85 | 10.73 |
| Descriptive captions | 1.28 | 2.49 | 10.16 |

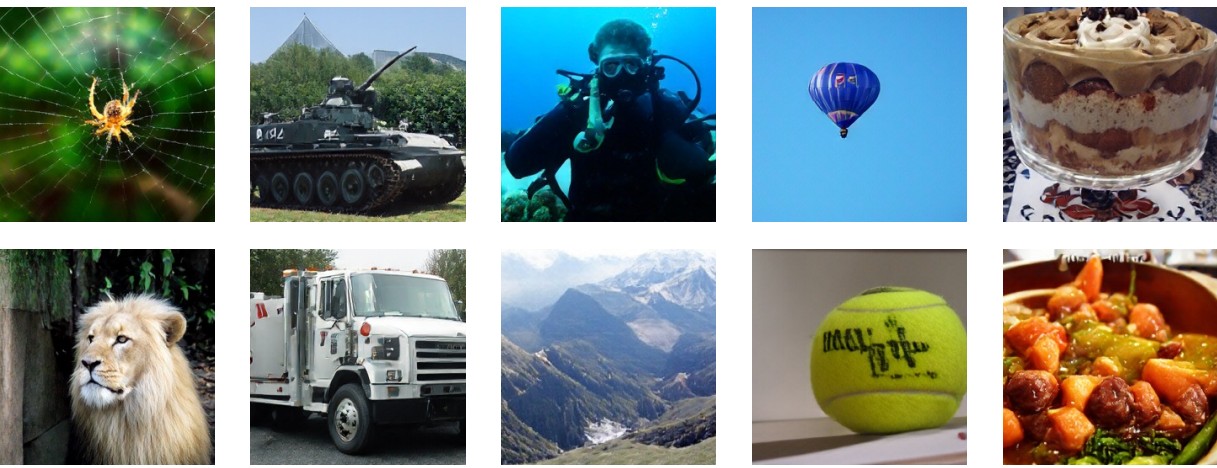

*Figure 8.* Additional C2I examples produced by ConceptTok-B-L-128 using the LlamaGen-XL generator.

## D. Limitations

First, ConceptTok relies on SAE-derived concept spaces from pre-trained vision-language models (VLMs). The coverage of the concept space is therefore inherited from the underlying VLM, its pre-training data, and the dataset used for SAE training. Different SAE choices and sparsity levels $K$ may influence the quality of semantic supervision and downstream performance. Second, our current concept loss is defined at the image level via average pooling, and a promising future direction is to develop patch-level or token-level concept supervision. Third, our tokenizer training uses ImageNet with templated captions. Scaling to larger and more diverse multimodal datasets may enable richer and more compositional semantic structures. Finally, while we focus on discrete tokenization, extending concept guidance to continuous tokenizers for diffusion-based generation remains an important future direction.

*"A man is playing with an elephant in a field."*     *"A cat is sitting next to a pumpkin and other vegetables."*

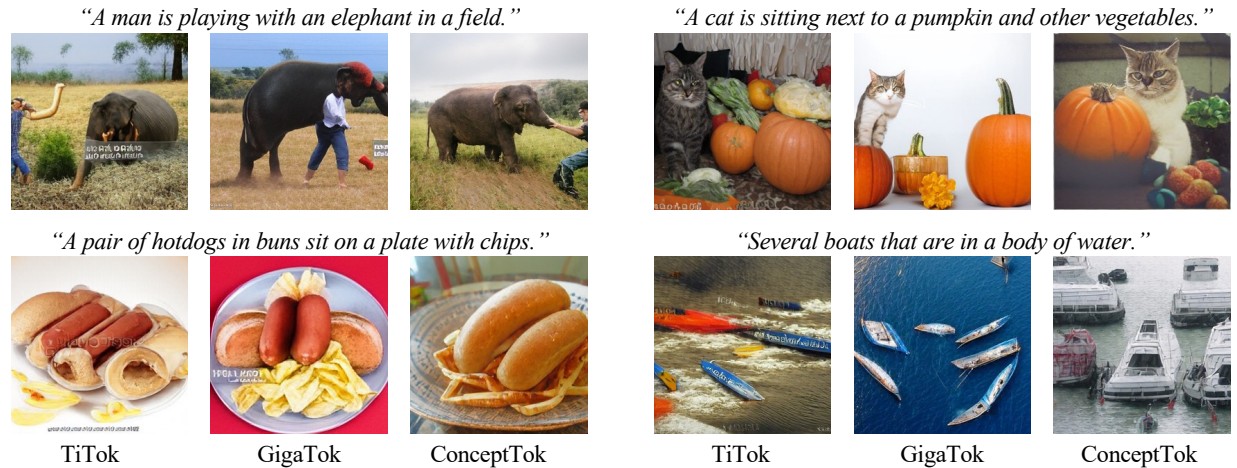

*"A pair of hotdogs in buns sit on a plate with chips."*     *"Several boats that are in a body of water."*

TiTok    GigaTok    ConceptTok      TiTok    GigaTok    ConceptTok

*Figure 9.* Additional T2I examples produced by different tokenizers using MaskGen-L.

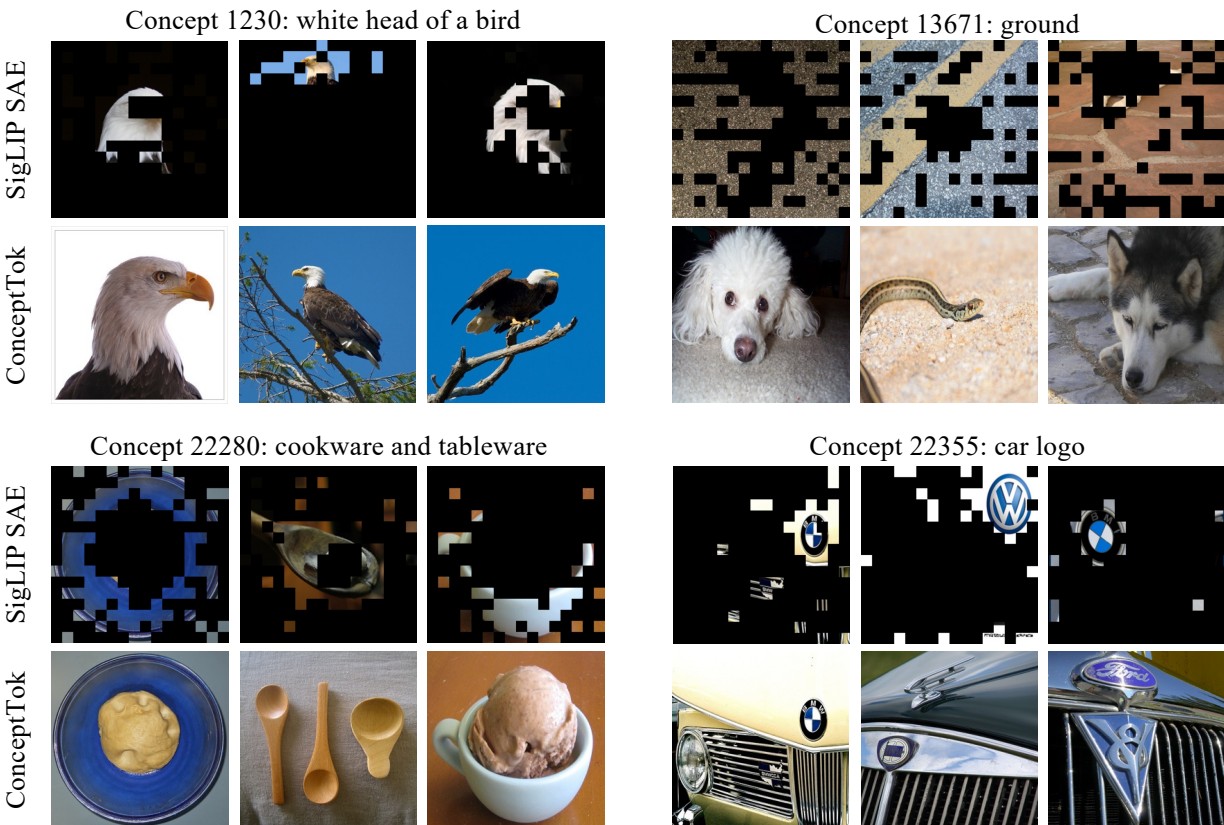

*Figure 10.* Additional qualitative results on concept alignment. For each concept index identified by the SigLIP SAE, we show patch-level activations highlighting regions that strongly activate the concept index, along with images retrieved by ConceptTok whose latent representations yield high scores for the same concept index.

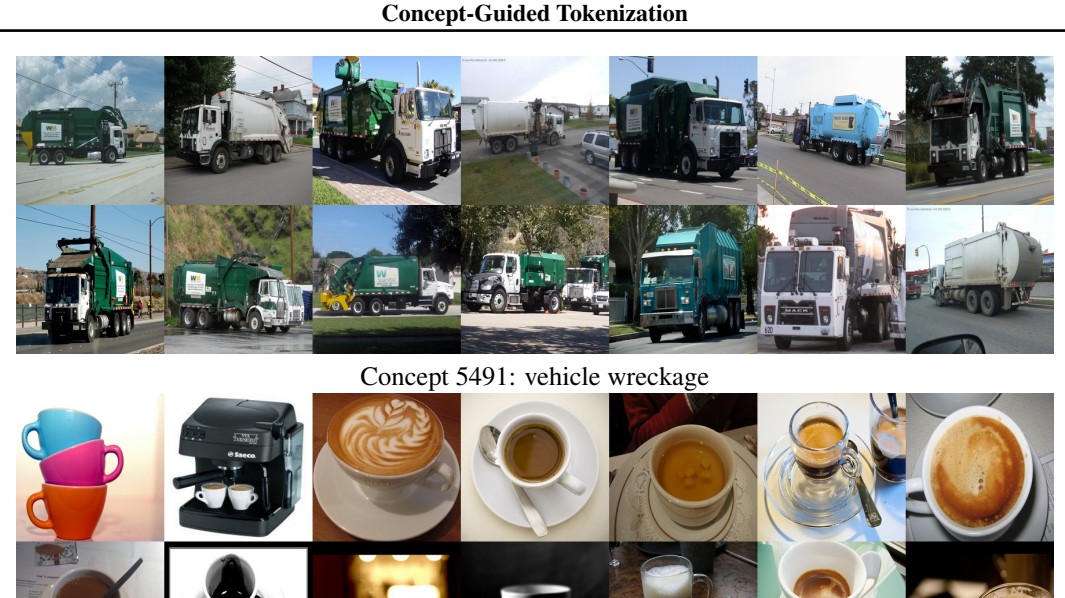

Concept 5491: vehicle wreckage

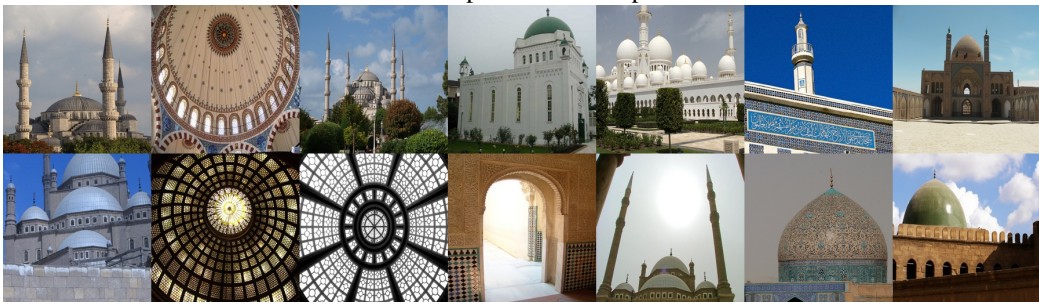

Concept 13586: teacup

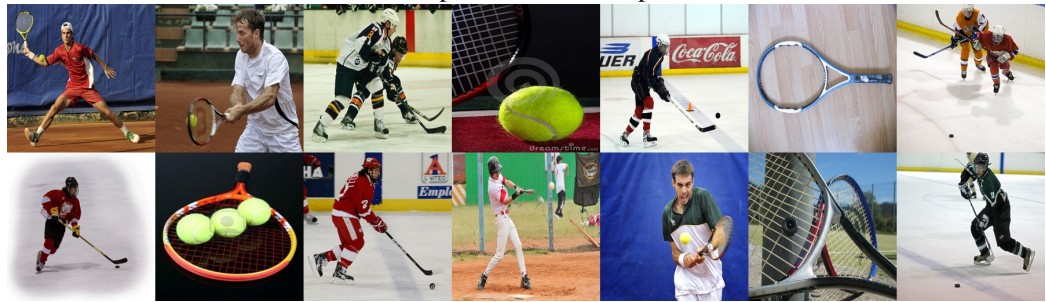

Concept 14617: church spire

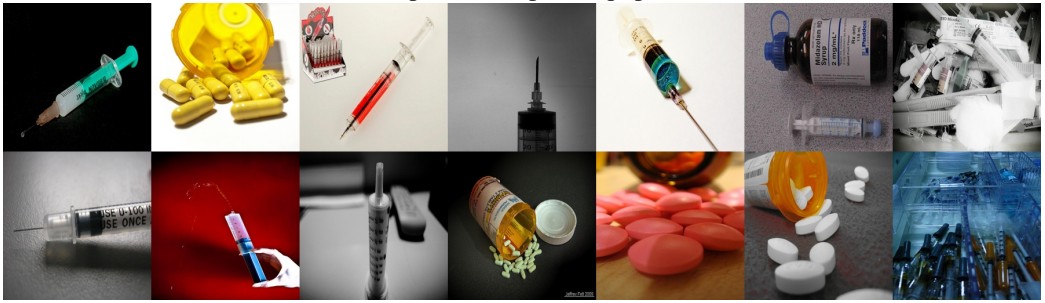

Concept 15343: sports equipment

Concept 20551: medical instrument

*Figure 11.* Representative images corresponding to selected SAE-derived concept indices.

