# OpenReview forum: "Concept-Guided Tokenization: Closing the Gap Between Reconstruction and Generation"
_ICML.cc/2026/Conference — ICML 2026 regular_

### Official Review · Reviewer_M82R · 2026-02-21

**Soundness:** 3
**Presentation:** 3
**Significance:** 2
**Originality:** 2
**Overall Recommendation:** 4
**Confidence:** 3

**Summary:**

This paper proposes ConceptTok, a novel image tokenizer that addresses the reconstruction-generation trade-off by integrating explicit semantic guidance. Unlike traditional methods that rely solely on pixel reconstruction or dense feature alignment, ConceptTok employs Sparse Autoencoders (SAEs) to project pre-trained vision-language model features into a disentangled semantic concept space. The tokenizer is trained to predict sparse concept indices from this space while jointly encoding image-text pairs.

**Compliance With Llm Reviewing Policy:**

Affirmed.

**Final Justification:**

While, like other reviewers, I still have concerns about the paper's novelty, I don't think it's a major issue. Secondly, could the authors have gone further to achieve sparsity? Because representations of other modalities still tend to have biases, but for example, ‘Gao Y, Chen K, Peng Z, et al. Knowledge Transfer from Interaction Learning[C]//Proceedings of the IEEE/CVF International Conference on Computer Vision. 2025: 3585-3595.’ attempted to achieve concept transfer in a sparser attention space. I hope there can be more discussion and expansion on this, therefore I can only raise the score to 4.

**Key Questions For Authors:**

See the summary

**Limitations:**

yes

**Strengths And Weaknesses:**

This is an interesting and potentially impactful piece of work, but several concerns should be addressed to strengthen the paper.
First, the notion of “concept” appears to be treated somewhat differently between text and image modalities, yet the definition—particularly for visual concepts—remains under-specified. While the authors provide both quantitative and qualitative analyses (which are commendable), the abstract nature of “concept” calls for a more rigorous theoretical foundation. I would encourage the authors to ground their formulation in established frameworks—e.g., from semiotics or cognitive science—to provide a clearer, formal characterization of what constitutes a “concept” in their setting. Existing literature has explored related priors; explicitly positioning this work against that background would enhance its conceptual clarity.

Second, Table 8 raises a concern regarding the contribution of the concept signal during training: a seemingly minimal intervention (e.g., a weight of 0.01) yields a substantial performance gain. This warrants further investigation. To better assess the role and efficacy of the proposed concept module, I recommend including convergence curves for both the main task loss and the concept-related loss throughout training. Such analysis would help validate whether the reported improvements genuinely stem from meaningful concept learning rather than incidental optimization effects.

Finally, while I acknowledge that I am not a domain expert in this specific subfield, I found it difficult to pinpoint the core novelty of the work beyond (1) aligning pre-trained model representations and (2) incrementally incorporating an additional modality. The latter, in particular, appears to overlap with existing approaches cited in the Related Work. The authors should more clearly articulate what fundamentally distinguishes their method from prior art and why their specific design choices are necessary or uniquely effective.

Overall, I believe the paper has merit, and I would be happy to raise my score if these points are adequately addressed. I wish the authors continued success in their research.

---

> ### Author Rebuttal · Authors · 2026-03-31
>
> We sincerely appreciate your constructive comments on improving our paper. We detail our response below point by point. Please kindly let us know if our response addresses the questions you had for this paper.
>
>
> ### [W1] Concept Definition
> We will follow the reviewer's valuable suggestion to clarify the definition of concepts in the revised Introduction.
> - We follow recent works [1][2], where a concept is defined as a latent representation shared by a set of samples (e.g., highly activated image patches). This definition aligns with the prototype and exemplar theory of concepts in cognitive science [3], where the latent representation corresponds to a prototype and the activating samples constitute its exemplars.
> - Concretely, a visual concept is operationalized as a latent factor in the SAE space of a VLM (representing a prototype), together with its top-activating image examples and corresponding textual descriptions (representing exemplars). Note that
>     - (1) concepts, in this case, are inherently multi-modal, as the latent representation aligns visual and textual modalities through semantic alignment learned during VLM pre-training.
>     - (2) concepts can exist at multiple levels of abstraction (e.g., legs vs. dogs), although our SAE concept space is flattened; this is consistent with cognitive science perspectives that "All concepts are either primitive or complex, and all complex concepts are defined in terms of primitive ones" [4].
>
> ### [W2] Convergence Curves
> We provide convergence curves for trade-off values 0.01 and 0.1 in Tab.8 (Fig.A in anonymous.4open.science/r/ICML-ConceptTok/Figure_Table.pdf). In both cases, the concept loss decreases steadily throughout training, while the reconstruction loss remains largely stable except for the expected transition around epoch 80, where GAN training is introduced. These trends suggest that concept alignment is learned consistently without destabilizing reconstruction optimization, supporting that the observed gains are associated with meaningful concept learning rather than incidental optimization effects.
>
> ### [W3] Novelty and Contribution
> We thank the reviewer for this helpful comment. Our core novelty is introducing concept guidance, rather than simply aligning to pretrained representations or adding an extra modality.
>
> - Instead of aligning tokenizer latents to holistic dense pretrained features, we align them in an SAE-derived sparse concept space. This is important because dense features are high-dimensional and semantically entangled, whereas sparse concept indices provide sharper, more structured supervision.
>
>   - (1) This is validated in Sec. 5.3: Comparison with Holistic Feature Alignment. Replacing our concept guidance with holistic high-dimensional cosine alignment yields weaker feature alignment (CKNNA 0.39 vs. 0.48) and worse generation (ImageNet gFID 4.84 vs. 4.13), showing that the gain comes from concept-space alignment, not feature alignment alone.
>   - (2) We further verify that the advantage of concept-space alignment is not limited to tokenizer training. In diffusion generator training [5], when all settings are kept identical except for the alignment way, concept-space alignment (CG) also outperforms REPA’s holistic dense alignment [5].
>     >    |Generator|Alignment weight|Iter.|gFID|
>     >    |-|-|-|-|
>     >    |SiT-L/2-REPA|0.5|400K|9.7|
>     >    |SiT-L/2-CG|0.1|400K|9.2|
>     >    |SiT-L/2-CG|0.2|400K|8.7|
>
> - The role of text provides complementary semantics for a structured latent space. Our ablation supports this design choice: replacing encoder-only text conditioning with decoder-only leads to worse 100-epoch ImageNet rFID with B-B-64 (5.5 vs.5.0), and will add the final 200-epoch results in the revision/discussion.
>
> [1] Mechanistic understanding and validation of large AI models with SemanticLens, Nature Machine Intelligence, 7:1572–1585, 2025.
>
> [2] Large Multi-modal Models Can Interpret Features in Large Multi-modal Models, ICCV 2025.
>
> [3] Concepts, Kinds, and Cognitive Development, Oxford University Press, 1992.
>
> [4] The Origin of Concepts, Oxford University Press, 2009.
>
> [5] Representation Alignment for Generation: Training Diffusion Transformers Is Easier Than You Think, ICLR 2025.

---

> > ### Author Rebuttal · Reviewer_M82R · 2026-04-02
> >
> > While, like other reviewers, I still have concerns about the paper's novelty, I don't think it's a major issue. Secondly, could the authors have gone further to achieve sparsity? Because representations of other modalities still tend to have biases, but for example, ‘Gao Y, Chen K, Peng Z, et al. Knowledge Transfer from Interaction Learning[C]//Proceedings of the IEEE/CVF International Conference on Computer Vision. 2025: 3585-3595.’ attempted to achieve concept transfer in a sparser attention space. I hope there can be more discussion and expansion on this, therefore I can only raise the score to 4.

---

> > > ### Author Response · Authors · 2026-04-08
> > >
> > > We sincerely appreciate the reviewer’s positive feedback and provide additional discussion below.
> > >
> > > ### [D1] Attention Space
> > > We thank the reviewer for this insightful suggestion and will discuss the work [1] in the revision.
> > >
> > > - In ConceptTok, direct attention-matrix alignment is not straightforward, because its encoder processes heterogeneous tokens (text, patch, and learnable latent tokens), whereas the SigLIP vision encoder contains only image tokens. As a result, the corresponding attention maps do not have a fully matched structure, making direct alignment less natural.
> > >
> > > - To nevertheless evaluate the reviewer’s suggestion, we apply attention-matrix alignment [1] in the REPA [2] setting, where both the diffusion Transformer image generator and the pre-trained model take images as input.
> > >   -  For fairness, we follow the original setting and keep all other training choices unchanged [2], replacing only the alignment objective with attention-matrix alignment, using the default trade-off weight of 1.0 as in [1].
> > >   -  The results show that attention alignment is indeed helpful in our setting and may further benefit from a more thorough search over the alignment weight, although it is currently less effective than REPA or our concept-guided alignment under the present setup.
> > >   -  We conjecture that this difference mainly stems from the different emphasis of the alignment spaces. Attention-based alignment naturally focuses on interaction patterns and salient semantic relations, which can be highly effective in settings such as classification and detection [1]. In contrast, the tokenizer and generator are for image generation oriented tasks, where the latent representation should preserve all visual detail for reconstruction and generation.
> > >
> > >     > |Generator|Alignment weight|Iter.|gFID|
> > >     >  |-|-|-|-|
> > >     >  |SiT-L/2|0.0|400K|18.8|
> > >     >  |SiT-L/2-Atten|1.0|400K|15.6|
> > >     >  |SiT-L/2-REPA|0.5|400K|9.7|
> > >     >  |SiT-L/2-CG|0.2|400K|8.7|
> > > -  We also consider interaction/attention space a promising alternative for concept extraction, and we will further investigate this direction in future work.
> > >
> > >
> > > [1] Knowledge Transfer from Interaction Learning, ICCV 2025.
> > >
> > > [2] Representation Alignment for Generation: Training Diffusion Transformers Is Easier Than You Think, ICLR 2025.

---

### Official Review · Reviewer_tzzU · 2026-03-10

**Soundness:** 2
**Presentation:** 3
**Significance:** 2
**Originality:** 2
**Overall Recommendation:** 3
**Confidence:** 5

**Summary:**

This paper proposes Concept-Guided Tokenization (ConceptTok), an image tokenizer that incorporates textual information into the encoder and introduces a concept-guided training objective in addition to the reconstruction loss. The goal is to encourage the model to learn more semantically rich latent image representations. The authors conduct experiments on ImageNet-256 for class-conditional image generation and COCO for text-to-image generation. The results show a certain level of improvement compared with the baseline.

**Compliance With Llm Reviewing Policy:**

Affirmed.

**Key Questions For Authors:**

Please refer to the weaknesses above.

Additionally, I have the following question:

- Why are the latent tokens average-pooled when computing the concept loss? This design choice is not entirely clear to me. If the latent tokens are expected to correspond to semantic concepts, one might expect some form of token-level alignment between the 1D tokens and the corresponding concepts, as suggested in “Highly Compressed Tokenizer Can Generate without Training” by Lucas Beyer et al. (ICML 2025). Simply averaging the tokens removes spatial and token-level distinctions, which could potentially eliminate this desirable property. It is therefore unclear why average pooling is an appropriate operation for computing the concept loss. I would appreciate the authors’ clarification on the motivation behind this design choice and whether alternative formulations (e.g., token-level alignment) were considered.

**Limitations:**

Please refer to weaknesses and key questions.

**Strengths And Weaknesses:**

Strengths:
- Image tokenization is an important research area, and reconstruction alone may not be a sufficient training objective. This paper attempts to address this limitation by incorporating semantic guidance into the tokenizer training.

- The paper is generally well written and easy to follow.

Weaknesses:
- Limited novelty. Using textual information to guide image tokenization has already been explored in prior work such as TexTok and TA-TiTok. The main architectural difference is relatively minor: TexTok incorporates text information in both the encoder and decoder, TA-TiTok incorporates it only in the decoder, while ConceptTok proposes incorporating it only in the encoder. In addition, the proposed concept-guided training objective can be viewed as another form of representation alignment, which has already been widely studied and shown to improve tokenizer convergence and generation performance.

- The main claims are not sufficiently supported by experiments. On the text incorporation side, the authors claim that incorporating text in the decoder leads to latent representations that remain weakly grounded in semantics and are less effective as self-contained representations for image generation models. However, no direct experimental evidence is provided to support this claim. Similarly, on the representation alignment side, the paper discusses two potential challenges of directly aligning with high-dimensional external model features, but these arguments are not validated experimentally.

- The experimental evaluation is insufficient to support the paper’s conclusions. Experiments should be designed to directly test the above claims. For example, a direct comparison with TA-TiTok-VQ is missing in both the ImageNet and COCO experiments. In addition, the text-to-image experiments are relatively limited and do not include comparisons with stronger or more recent state-of-the-art models, making it difficult to assess the scalability and practical benefits of the proposed ConceptTok.

- Potential concerns about the concept targets. The concept targets are extracted by running a sparse autoencoder on pre-trained models. It is unclear whether such extracted concepts provide sufficient coverage or generalizability. The approach may introduce limitations in the semantic space, and the paper does not provide a clear method to evaluate or validate this aspect.

---

> ### Author Rebuttal · Authors · 2026-03-31
>
> We sincerely appreciate your constructive comments and the opportunity to clarify the scope of our contribution and provide additional evidence.
>
> ### [W1&W2] Novelty and Validation
> We would like to clarify the main contributions with clearer evidence.
> - Our representation alignment is not simply another instance of standard dense feature alignment. Instead of a holistic-aligning tokenizer, we align the latents to pretrained features in an SAE-derived sparse concept space. This distinction is important because dense pretrained features are both high-dimensional and semantically entangled, whereas sparse concept indices provide sharper and more structured supervision.
>   - (1) This is validated in Sec. 5.3 *Comparison with Holistic Feature Alignment*. Replacing our concept guidance with holistic high-dimensional cosine alignment yields weaker feature alignment (CKNNA 0.39 vs. 0.48) and worse generation (ImageNet gFID 4.84 vs. 4.13), showing that the gain comes from concept-space alignment, not feature alignment alone.
>   - (2) We further evaluate the learned tokenizer representations through ImageNet linear probing, following TiTok. To avoid class information leakage from the text prompt, we use the generic prompt “A photo” for ConceptTok-B-L-128†. Under this setting, ConceptTok achieves **0.66** accuracy, outperforming TiTok-L-L-128 (**0.54**) and GigaTok-B-L-256 (**0.61**).
> -  We agree that prior work, such as TexTok and TA-TiTok, demonstrates the effectiveness of decoder-side text conditioning and the combination of both sides.
>    - (1) Our point, however, is not that decoder-side conditioning is ineffective, but that it may primarily improve reconstruction through the detokenizer, whereas our objective is to make the latent representation itself more semantically grounded and self-contained for downstream tasks. This motivation extends beyond image generation to other image-conditioned generation tasks raised by Reviewer GF3A, which we are investigating.
>    - (2) To examine this distinction directly, we compare encoder-only and decoder-only text conditioning in our ablation. Under the B-B-64 setting on ImageNet at 100 epochs, encoder-only conditioning performs better than decoder-only conditioning (rFID 5.0 vs. 5.5).
> ### [W3] More Comparisons
> Thank you for this suggestion. We agree that our claims should be supported by as direct an experimental evaluation as possible, beyond the comparison with GigaTok (ICCV 2025) included in the paper.
> - However, a controlled comparison with TA-TiTok-VQ is challenging because it is trained on DataComp with recaptioned data, and neither an ImageNet-trained checkpoint nor the corresponding ImageNet caption setup has been publicly released. Likewise, TexTok does not provide public checkpoints or code, which prevents direct reproduction.
> - To further strengthen the empirical evidence, we followed the reviewer’s suggestion and extended our concept-space alignment to a stronger diffusion generative model, REPA [1]. Under identical settings on SiT-L/2, with 400K training iterations and the same pre-trained model, concept-space alignment improves gFID from **9.7 to 8.7** relative to REPA’s holistic dense alignment. This suggests that the advantage of concept-space alignment extends beyond tokenizer training and remains beneficial in stronger diffusion generative settings.
> ### [W4] Concept Generalization
> - We would like to clarify that the SAE-derived concept space is obtained through a sparse decomposition of the feature space of a strong pre-trained vision-language model. In this sense, its semantic coverage is inherited from the underlying pre-trained model and its pre-training data, while the SAE primarily provides a sparser and more disentangled representation for supervision.
> - Please also refer to our empirical validations in response to [W1&2] in Reviewer GF3A.
> ### [Q1] AvgPooling
> - Our current concept supervision is defined at the image level rather than the token level: the target is the set of SAE concept indices activated for the whole image. We therefore use average pooling to obtain a global latent summary whose granularity matches that of the supervision.
> - This averaging does not explicitly overwrite token-specific roles. Indeed, our token-level analysis suggests that specialization still emerges: for concept 1230 in Fig. 10, all top-20 highest-activating images contain latent token 148 at position 20, and 16/20 contain latent token 1945 at position 40, with similar recurring patterns for other concepts. This suggests that average pooling mainly aggregates global concept evidence, while token-level specialization can still emerge.
> - We agree that token-level alignment is a promising direction. However, in our current setting, the latent tokens lack explicit token-level semantic labels, making it difficult to reliably formulate such alignment.
>
> [1] Representation Alignment for Generation: Training Diffusion Transformers Is Easier Than You Think, ICLR 2025.

---

> > ### Author Rebuttal · Reviewer_tzzU · 2026-04-03
> >
> > I thank the authors for their rebuttal, which partially addresses my concerns. However, I believe a direct comparison to TA-TiTok-VQ-128 is necessary, as both methods use a ViT-Base encoder and a ViT-Large decoder, making them closely comparable in architecture.
> >
> > While TA-TiTok is trained on recaptioned DataComp, it is not immediately clear that this provides a decisive advantage over standard benchmarks such as ImageNet or COCO. Given that many tokenizer comparisons in the paper are not strictly apple-to-apple and instead aim to provide a general sense of performance, including a more controlled and direct comparison would significantly strengthen the evaluation.

---

> > > ### Author Response · Authors · 2026-04-08
> > >
> > > We sincerely appreciate the constructive suggestion and agree that a direct comparison with TA-TiTok would be valuable.
> > >
> > > ### [W3&D1] Controlled Comparison with TA-TiTok
> > >
> > > - T2I Comparison in Tab. 2
> > >
> > >   To provide a more direct comparison, we additionally evaluate TA-TiTok-B-L-128 under a setting aligned with Tab. 2, with reconstruction metrics computed in the same manner. For T2I generation, due to limited rebuttal time, we report MaskGen results trained for 50 epochs rather than the full training schedule used in Tab. 2. Therefore, for fairness, we report 50-epoch MaskGen results for both TA-TiTok and ConceptTok, as shown in the table below.
> > >   - TA-TiTok, which is trained on recaptioned DataComp, achieves better reconstruction on COCO, whereas our method, trained on ImageNet, achieves better reconstruction on ImageNet.
> > >   - When informative captions are unavailable and replaced with a generic prompt such as “A photo,” TA-TiTok exhibits a larger drop in ImageNet reconstruction. This indicates that TA-TiTok relies more heavily on caption information at reconstruction time, whereas our tokenizer is more robust when such captions are unavailable.
> > >   - The controlled comparison provides further evidence that ConceptTok achieves stronger downstream generation performance under the same 50-epoch CC3M T2I training setting.
> > >
> > >     |  |ImageNet (w caption) rFID ↓| ImageNet("A photo.") rFID ↓|COCO rFID ↓|50-epoch COCO T2I gFID ↓|
> > >     |-|-|-|-|-|
> > >     TA-TiTok-B-L-128|1.53|2.64|2.43|16.54|
> > >     |ConceptTok-B-L-128†|1.39|1.95|2.85|12.64|
> > >
> > > - Image Inpainting as a Representative **Image-Conditioned Generation** Task
> > >   - Following reviewer GF3A’s suggestion, we train an inpainting ViT on CC3M to take the tokenizer tokens of a masked image as input and predict the tokenizer tokens corresponding to the original unmasked image, using a cross-entropy loss.
> > >   - We evaluate on COCO using a fixed center mask covering 40% of the image width and height, and report the LPIPS and SSIM of the inpainted image against the tokenizer reconstruction of the original image.
> > >   - The results show that our tokenizer is more effective for image-conditioned generation tasks, which we attribute to its more self-contained latent representations. Such representations are better suited for downstream image-conditioned generation, as they preserve semantic information more directly in the latent tokens.
> > >
> > >     |  |LPIPS↓|SSIM↑|
> > >     |-|-|-|
> > >     |ConceptTok-B-L-128†|0.210|0.847|
> > >     |TA-TiTok-B-L-128|0.264|0.829|

---

### Official Review · Reviewer_1ZGV · 2026-03-13

**Soundness:** 2
**Presentation:** 3
**Significance:** 3
**Originality:** 3
**Overall Recommendation:** 4
**Confidence:** 4

**Summary:**

This paper proposes a new tokenization framework called ConceptTok to address
the trade-off problem where existing image tokenizers perform well in
reconstruction but poorly in generation. This method integrates text into the
encoder, enabling the tokenizer to receive both visual and linguistic information as
input. Furthermore, it decomposes the features of pre-trained SigLIP into a
semantic concept space using TopK SAE and utilizes this as a concept guidance.
Experimental results demonstrate strong performance on ImageNet and COCO-30k
for both C2I and T2I tasks, mitigating the gap between reconstruction and
generation.

**Compliance With Llm Reviewing Policy:**

Affirmed.

**Final Justification:**

The additional analyses and clarifications provided in the rebuttal have addressed my concerns.
In particular, I appreciate the added ablation results, along with the further clarification regarding the trade-off perspective and the comparison with TiTok.
Based on these updates, I consider my concerns to be largely resolved and will increase my score to 4.

**Key Questions For Authors:**

1. Are there expected to be further performance improvements with the use of
more advanced models like SigLIP2?
2. Since the method requires first training the SAE and then freezing it to guide
tokenizer training, a discussion of the additional training cost introduced by this
extra stage would be valuable.
3. Although RAE [2] is not a direct tokenizer baseline, it would be valuable to
discuss how ConceptTok relates to recent concurrent work such as RAE, which
offers a strong alternative design for generative latent spaces.

[2] Zheng et al., Diffusion Transformers with Representation Autoencoders. arXiv
2025

**Limitations:**

yes

**Strengths And Weaknesses:**

Strengths
- To validate the proposed method, the paper provides a diverse set of quantitative
metrics and qualitative analyses, including reconstruction and generation
evaluations (e.g., rFID and gFID), results on both ImageNet and COCO-30k, and
concept alignment visualizations.
- It enhances persuasiveness by explaining the limitations of existing approaches
that align visual representations of pre-trained models with latent features, and by
describing how the proposed method differentiates itself.
- It improves understanding by clearly explaining the two techniques: the
text-integrated encoder and the concept-guided training objective.

Weaknesses
- Table 1 is difficult to interpret as a fair comparison because the compared
methods differ in generator architecture and parameter count. Although
ConceptTok uses only 128 latent tokens, this alone does not necessarily imply a
more efficient model, since its parameter count is still higher than several
baselines. Also, since the method is built upon TiTok, the advantage of using 128
tokens feels less clear given that TiTok itself can use even fewer latent tokens.
- TA-TiTok [1] suggests that decoder-side conditioning alone may already capture
high-level semantic information. In this context, the motivation for integrating text
tokens into the encoder would be more convincing if the paper included a direct
comparison with decoder-side text conditioning.
- Some of the supporting ablations are conducted under different tokenizer scales
from the main B-L setting used in the primary results, which makes the overall
empirical argument feel somewhat indirect.
- Minor presentation issue: It would be good to keep the presentation order of
ConceptTok in the introduction consistent with the other sections to improve the
overall consistency of the paper.


[1] Kim et al., Democratizing text-to-image masked generative models with compact
text-aware one-dimensional tokens. ICCV 2025

---

> ### Author Rebuttal · Authors · 2026-03-31
>
> We sincerely appreciate your comments on our paper. We detail our response below point by point. If you have any further concerns, we would be grateful if you could let us know.
>
> ### [W1] Comparison Fairness in Tab. 1
> Please kindly refer to our response to **[W3] in Reviewer GF3A** .
>
> ### [W2] Ablation of Decoder-only
> Thanks for the constructive comments.
> - Our point is not that decoder-side conditioning is ineffective. Rather, it may primarily improve reconstruction through the detokenizer, whereas our goal is to make the latent representation itself more semantically grounded and self-contained for downstream use, including settings where a more self-contained latent space may be beneficial. This motivation extends beyond image generation to other image-conditioned generation tasks raised by Reviewer GF3A.
>
> - Our current ablation also provides preliminary evidence in favor of encoder-only conditioning: under the B-B-64 setting on ImageNet at 100 epochs, decoder-only text conditioning yields worse reconstruction performance than encoder-only conditioning (ImageNet rFID 5.5 vs. 5.0). We will add the final 200-epoch results in the revision/discussion for completeness.
>
> ###  [W3] Ablation using B-L
> We thank the reviewer for pointing this out. Due to the limited rebuttal time, we were unable to complete the full B-L training runs, but we could provide some intermediate B-L results in the discussion period.
>
> ###  [W4] Presentation Order
> Thank you for pointing this out. We will revise the paper to maintain consistent ordering.
>
> ###  [Q1] Stronger Pre-trained Models
> Thank you for this question. While stronger pre-trained models may in principle provide better concept supervision, our preliminary results suggest that the gain is not automatic. Specifically, replacing SigLIP with SigLIP2 under the same B-B-64 setup yields a slightly worse ImageNet reconstruction FID at 120 epochs (5.27 vs. 4.93). This suggests that the benefit is not automatic under the current setup. One possible reason is that a stronger backbone may induce a differently distributed concept space, requiring separate tuning of the alignment hyperparameters.
>
> ###  [Q2] SAE Cost
> Thank you for raising this important point. We would like to clarify that the additional cost introduced by the SAE is modest.
> - The SAE is trained once as an auxiliary interpretability module, and the resulting concept space can be reused across tokenizer training runs and other model steering tasks [1].
> - SAE training is substantially cheaper than tokenizer training: it requires only 1.35 hours, whereas training the B-L tokenizer takes roughly 200 hours on 8 A100 GPUs.
>
> ###  [Q3]  Relation to RAE
> We agree that RAE is an important concurrent line of work and appreciate the suggestion to discuss it.
> - Both RAE and our method aim to build more semantically meaningful latent spaces: RAE relies on fixed pretrained representation encoders, whereas our method uses concept-level feature alignment.
> - Yet, our intuition aligns with PS-VAE [2], which suggests that using a fixed encoder directly may be insufficient and that fine-tuning the encoder with feature alignment and reconstruction objectives remains important. From this perspective, ConceptTok is complementary to these approaches and provides a more structured concept-level alignment objective.
>
> [1] Sparse Autoencoders Reveal Selective Remapping of Visual Concepts during Adaptation, ICLR 2025.
>
> [2] Both Semantics and Reconstruction Matter: Making Representation Encoders Ready for Text-to-Image Generation and Editing, arXiv:2512.17909, 2025.

---

> > ### Author Rebuttal · Reviewer_1ZGV · 2026-04-04
> >
> > Thank you for providing the additional results on SigLIP2 and the comparison with decoder-only conditioning. I also appreciate the clarification regarding the SAE cost and the discussion of the relation to RAE. These additions address a substantial portion of my earlier concerns.
> >
> > However, my concern regarding Table 1 remains only partially resolved. I understand the authors' point that GigaTok is the most recent and closely related baseline. That said, compared with GigaTok, ConceptTok appears to have worse reconstruction performance while only improving generation performance. Could the authors clarify under what criteria this is considered a better overall trade-off?
> > In addition, since ConceptTok is built upon TiTok, it is still not entirely clear from Table 1 in what aspects it improves over TiTok.

---

> > > ### Author Response · Authors · 2026-04-08
> > >
> > > We sincerely thank the reviewer for the insightful follow-up. Below, we clarify the remaining points and provide additional comparisons.
> > >
> > > ### Follow-up on Previously Discussed Ablations [W2] & [W3].
> > > As a follow-up to our first-round response, we also include the updated results for the two ablations that were previously discussed but not yet complete at rebuttal, for completeness.
> > > - [W2] B-B-Deocder Only
> > >     -  Continued to the previous results, we now provide a 200-epoch comparison on ImageNet under B-B-64: decoder-only text conditioning gives a worse ImageNet rFID than encoder-only conditioning (5.39 vs. 3.84). This supports our design choice of using text conditioning in the encoder.
> > >
> > > - [W3] Ablation using B-L
> > >   - We now provide a same-scale preliminary ablation under the main B-L-128 setting. After 50 epochs of training, ConceptTok-B-L-128 achieves an ImageNet rFID of 2.59, while the variant without concept guidance obtains 2.70. Although these are still intermediate results rather than full runs, they already show a consistent advantage of concept guidance under the same tokenizer scale, supporting the trend observed in our ablations.
> > >
> > > ### [D1] Clarification on the Trade-off
> > > We would like to clarify the criterion under which we consider the trade-off favorable:
> > >
> > > - The tokenizers compared in Tab. 1 are not intended for image compression, but for supporting downstream tasks such as image generation.
> > > - In this setting, reconstruction primarily evaluates how well the tokenizer fits the training distribution, whereas downstream generation requires more generalizable and transferable representations beyond reconstruction fidelity.
> > > - From this perspective, we view ConceptTok’s advantage as **generation-oriented** rather than uniformly superior on every metric. Compared with GigaTok, ConceptTok shows slightly worse reconstruction performance but stronger downstream generation under a shorter latent sequence, as shown in Tabs. 1 and 2, which we view as **a more favorable trade-off for generative tokenization**.
> > >
> > > ### [D2&W1] Comparison with TiTok
> > > We understand and appreciate the reviewer’s concern regarding the comparison with TiTok in Tab. 1.
> > > - In Tab. 1, a direct comparison is less controlled because the original TiTok results are reported with different token lengths and different image generators.
> > > - We therefore refer to **the T2I comparison with TiTok-B-L-128 in Tab. 2**, using the checkpoint released in the follow-up work TA-TiTok by the same group as TiTok. In this comparison, the token length and image generator are matched, and the overall model sizes are also broadly comparable, since our method only adds a frozen pre-trained text encoder. Under this controlled setting and comparable parameter, ConceptTok achieves better T2I generation performance than TiTok, indicating that its main improvement over TiTok lies in downstream generation quality.
> > > - Note that we also compare with **its follow-up work TA-TiTok**. Under a controlled setting with the same token length, image generator, and model size, ConceptTok achieves better downstream generation performance, including lower 50-epoch COCO T2I gFID↓ (12.64 vs. 16.54) and better image inpainting performance in terms of LPIPS↓ (0.210 vs. 0.264). This indicates that the main improvement of ConceptTok over TA-TiTok lies in downstream generation quality. Please refer to **[W3&D1]** for a more detailed discussion in our response to reviewer **tzzU**.

---

### Official Review · Reviewer_GF3A · 2026-03-13

**Soundness:** 3
**Presentation:** 4
**Significance:** 4
**Originality:** 3
**Overall Recommendation:** 5
**Confidence:** 3

**Summary:**

In this work, the authors learn a conceptually aligned visual tokenizer. They achieve this by first aligning the tokenized representation with a pretrained visual representation (SigLIP) that was sparsified using an SAE and a top-K filtering. Secondly, they also include textual captions that encode the class of the the image being tokenized. The token is then quantized and decoded in a manner similar to TiTok. Their experimental results indicate that the method is competitive but not state of the art for reconstruction, but has stronger results in Text-to-Image and Class-to-Image generation. They provide informative ablation experiments and analyze the latent space organiztion of their learned tokenizations.

**Compliance With Llm Reviewing Policy:**

Affirmed.

**Final Justification:**

The authors' answer my questions, in particular any scaling concerns of their method.

**Key Questions For Authors:**

* What happens when text captions are not available? Would this become a limitation when needing to tokenize inputs for image-conditioned generation tasks?
* How does the method perform when using more dense and less class-aligned text captions, either from training on a different dataset or using a VLM to caption the training data?

**Limitations:**

Yes

**Strengths And Weaknesses:**

- Strengths
    - Exploring visual tokenizers, particularly those that are useful for visual reconstruction but preserve a notion of semantic concepts is interesting for generation and other tasks like world modelling.
    - The choices of conceptual alignment and textual alignment each seem to contribute to learning better tokenizations that are well-supported via ablation experiments.
    - The use of an SAE to distill salient concepts is an interesting innovation that appears to be effective.
    - The paper is well-written, clear and easy to understand.
- Weakness
    - Making use of class information, like in text, and evaluating on C2I generation makes the result a little less convincing.
    - I am interested in how these tokenizations generalize to unseen concepts. By utilizing class and textual information, do the tokens “overfit” to the ImageNet concept space and have a difficult time generating OOD concepts?
    - It is difficult to reconcile some design differences between methods, notable the performance gap arising from using 128 vs 256 tokens.
    - It would be helpful if there are any baselines that attempt to directly align with high-dimensional representations without using an SAE.

---

> ### Author Rebuttal · Authors · 2026-03-31
>
> We sincerely appreciate your comments on our paper. You may find our responses below for your concerns. Please kindly let us know if our response addresses the issues you raised in this paper.
>
> ### [W1&2] Generalization beyond C2I/ImageNet
> We would like to clarify that our results suggest generalization beyond C2I.
> - In Tab. 2, ConceptTok achieves the best T2I performance on COCO (gFID 10.73), supporting transfer to a more open-ended T2I setting.
> - In Tab. 3, ConceptTok also maintains strong concept alignment on COCO (F1 0.542), indicating that the learned semantic structure is not restricted to ImageNet concepts.
>
> ### [W3] Comparison Fairness
> We agree that Tab. 1 is not a fully controlled apples-to-apples comparison, since tokenizer evaluation depends on multiple coupled factors, including tokenizer scale, latent token length, training setup, generator family and generator scale.
>
> To make the comparison as fair as possible, we emphasize GigaTok as the most recent and closely related baseline, since it also incorporates semantic guidance and is evaluated with the same generator. Although some design differences still remain, only two factors are important for interpreting the comparison fairly:
>
> - The number of latent tokens matters: as shown in Tab. 4 of TiTok and Tab. 6 of ours, longer token sequences improve performance, but also incur higher inference cost.
> - Model size also matters: as shown in Tab. 10 of GigaTok and Tab. 7 of ours, larger tokenizers usually perform better, although the gains diminish at larger scales.
>
> With this context, we emphasize that ConceptTok achieves a stronger quality-efficiency trade-off. Compared with GigaTok-B-L-256, ConceptTok-B-L-128 uses fewer parameters and a shorter token sequence, yet achieves better C2I performance in Tab. 1 and better T2I performance in Tab. 2.
>
> ### [W4] Alignment Method Comparison
> Thanks for the suggestion.
> - GigaTok is a representative holistic high-dimensional alignment baseline without SAE, and ConceptTok outperforms it in both C2I and T2I generation.
> - Sec. 5.3 *Comparison with Holistic Feature Alignment* includes an explicit ablation that replaces our concept guidance with holistic high-dimensional cosine alignment. This variant yields weaker feature alignment (CKNNA 0.39 vs. 0.48) and worse generation quality (ImageNet gFID 4.84 vs. 4.13), showing that alignment in the SAE-derived concept space is more effective than direct dense-feature alignment.
>
> ### [Q1] Without Captions
> Thank you for this question.
> - This does not appear to be a major issue for the C2I/T2I inference settings, since they use only the tokenizer decoder at inference time.
> - When captions are unavailable during tokenization, a generic prompt such as “a photo” still works reasonably well: on ConceptTok-B-L-128†, the ImageNet rFID changes from 1.39 to 1.95. This preliminary result suggests some robustness to missing detailed captions.
> - Due to time limits, we are currently extending our tokenizer to image-conditioned tasks to further assess its effectiveness.
>
> ### [Q2] Dense & Less Class-aligned Captions
> We follow the reviewer’s suggestion and train ConceptTok using the BLIP-2-recaptioned "visual-layer/imagenet-1k-vl-enriched" dataset from HuggingFace. Compared with ConceptTok-B-L-128†, the recaptioned ConceptTok-B-L-128† achieves consistent gains in both reconstruction and generation, suggesting that ConceptTok can further benefit from denser captions.
> |  |ImageNet rFID ↓|COCO rFID ↓|COCO T2I gFID ↓|
> |-|-|-|-|
> |ConceptTok-B-L-128†|1.39|2.85|10.73|
> |Recaptioned ConceptTok-B-L-128†|1.28|2.49|10.16|

---

> > ### Author Rebuttal · Reviewer_GF3A · 2026-04-03
> >
> > - The improvement on COCO addresses my concern. However, are there comparisons to other methods in Table 3?
> > - Thank you for this clarification. In particular the different data scales makes sense, however it does raise the question of how this method generalizes (discussed further as Q1)
> > - In general I am happy with the authors’ responses. Given the response to W3 and Q1 which highlight the importance of scale, I was concerned about the scalability of this method when auxiliary information is not available. While prompting with “a photo” seems to worse performance quite substantially, making use of recaptioned data seems to perform quite well and may serve as a promising answers to my concerns.
> >
> > I’ll update my score to a 5.

---

> > > ### Author Response · Authors · 2026-04-08
> > >
> > > We thank the reviewer for the positive feedback and are glad that our rebuttal addressed some of the concerns. Below, we respond to the remaining concerns.
> > >
> > > ### [D1] More Results in Tab.3
> > > Thanks for the question. We did not include other methods in Tab. 3 because prior tokenizers do not perform SAE-based concept alignment, so their concept-level alignment scores are not directly computable under our metric.
> > >
> > > ### [D2&Q1] Image-conditioned Generation Tasks
> > > We appreciate the reviewer's suggestion. Following the reviewer's suggestion, we further evaluate our tokenizer on an image inpainting task to examine the image-conditioned generation setting and to test its robustness when informative captions are unavailable.
> > >
> > > - We train an inpainting ViT on CC3M to take the tokenizer tokens of a masked image as input and predict the tokenizer tokens corresponding to the original unmasked image, using a cross-entropy loss.
> > > - We evaluate on COCO using a fixed center mask covering 40% of the image width and height, and report the LPIPS and SSIM of the inpainted image against the tokenizer reconstruction of the original image.
> > > - The results show that our tokenizer remains effective for image-conditioned generation even when informative captions are unavailable. In particular, replacing the dataset caption with a generic prompt such as “A photo.” causes only a modest drop, while our method still outperforms TA-TiTok.
> > >
> > > |  |Caption|LPIPS↓|SSIM↑|
> > > |-|-|-|-|
> > > |ConceptTok-B-L-128†|dataset|0.210|0.847|
> > > |ConceptTok-B-L-128†|"A photo."|0.223|0.841|
> > > |TA-TiTok-B-L-128|dataset|0.264|0.829|

---

### Decision · Program_Chairs · 2026-04-30

**Decision:**

Accept (regular)

**Comment:**

The paper received mixed ratings from four reviewers, with two weak accepts, one accept, and one weak reject. Most of the reviewers acknowledged the contribution of the paper and the comprehensive experimental evaluation. They also felt that the contribution is meaningful to the community. One reviewer had concerns regarding the presentation clarity and some missing ablation studies. The rebuttal has made significant efforts to address those issues. Based on the overall score distribution and the rebuttal provided by the authors, AC decided to recommend a weak accept for this submission.